# Extracellular vesicles from symbiotic vaginal lactobacilli inhibit HIV-1 infection of human tissues

Rogers A. Ñahui Palomino [1], Christophe Vanpouille[1], Luca Laghi [2], Carola Parolin [3], Kamran Melikov[4], Peter Backlund[5], Beatrice Vitali[3] & Leonid Margolis[1]*

The vaginal microbiota, dominated by *Lactobacillus* spp., plays a key role in preventing HIV-1 transmission. Here, we investigate whether the anti-HIV effect of lactobacilli is mediated by extracellular vesicles (EVs) released by these bacteria. Human cervico-vaginal and tonsillar tissues ex vivo, and cell lines were infected with HIV-1 and treated with EVs released by lactobacilli isolated from vaginas of healthy women. EVs released by *L. crispatus* BC3 and *L. gasseri* BC12 protect tissues ex vivo and isolated cells from HIV-1 infection. This protection is associated with a decrease of viral attachment to target cells and viral entry due to diminished exposure of Env that mediates virus-cell interactions. Inhibition of HIV-1 infection is associated with the presence in EVs of several proteins and metabolites. Our findings demonstrate that the protective effect of *Lactobacillus* against HIV-1 is, in part, mediated by EVs released by these symbiotic bacteria. If confirmed in vivo, this finding may lead to new strategies to prevent male-to-female sexual HIV-1 transmission.

[1] Section on Intercellular Interaction, Eunice Kennedy Shriver National Institute of Child Health and Human Development, National Institutes of Health, Bethesda, MD 20814, USA. [2] Center of Foodomics, Department of Agro-Food Science and Technology, University of Bologna, Cesena 47123, Italy. [3] Department of Pharmacy and Biotechnology, University of Bologna, Bologna 40126, Italy. [4] Section on Membrane Biology, Laboratory of Cellular and Molecular Biophysics, Eunice Kennedy Shriver National Institute of Child Health and Human Development, National Institutes of Health, Bethesda, MD 20814, USA. [5] Biomedical Mass Spectrometry Core Facility, Eunice Kennedy Shriver National Institute of Child Health and Human Development, National Institutes of Health, Bethesda, MD 20814, USA. *email: margolis@helix.nih.gov

The vaginal microbiota, dominated mostly by *Lactobacillus* spp.[1], plays a key role in defending the female genital tract against numerous urogenital pathogens, including HIV-1[2–8]. In spite of the importance of this phenomenon, its mechanism remains largely unknown. Several mechanisms of *Lactobacillus*-mediated protection of vaginal cells from HIV-1 infection have been suggested by others[2,9–11] and by us[12]. These include inactivation of HIV-1 through acidification of the vagina to pH~4.5, capture of virions through bacterial membrane lectins, stimulation of anti-HIV-1 immune responses, and inhibition of the growth of pathogens responsible for bacterial vaginosis, which increase the risk of HIV-1 infection[13].

Recently, it was found that *Lactobacillus*, along with other bacterial and mammalian cells, releases extracellular membrane vesicles. Extracellular vesicles (EVs) are implicated in cell–cell communications in various biological systems[14–16]. Cell–cell interactions in general, and microbiota–host-cell crosstalk in particular, are multifactorial phenomena that involve complex contact interactions between cells and cellular factors; many of these cellular factors such as lipids, proteins, nucleic acids, polysaccharides, and various soluble factors are associated with EVs. Noticeably, until the last decade it was believed that Gram-positive bacteria were not capable of releasing EVs because of their thick cell walls. However, Lee et al.[17] showed that *Staphylococcus aureus* releases EVs[17]. In subsequent studies, EVs have been also isolated from *Lactobacillus* strains[18–24].

Although the role of Gram-positive bacterial EVs has been less studied than that of mammalian EVs or even Gram-negative bacteria, it was found that Gram-positive-derived EVs from *Lactobacillus* can stimulate the host immune and nervous systems[18], enhance the host immune responses against other bacteria[24], and induce cell apoptosis in hepatic cancer cells[19].

We hypothesized that EVs released by lactobacilli contribute to the *Lactobacillus*-mediated protection of vaginal viral infection. This protection can be simulated ex vivo, as we demonstrated earlier: several strains of vaginal *Lactobacillus* inhibited HIV-1 infection in human cervico-vaginal and tonsillar tissues ex vivo[12]. Human tissues ex vivo retain tissue cytoarchitecture and provide an adequate experimental model to study the pathogenesis of various human viruses[25], in particular HIV-1.

Here, we investigate whether EVs derived from four different strains of *Lactobacillus* (*L. crispatus* BC3, *L. crispatus* BC5, *L. gasseri* BC12, and *L. gasseri* BC13) isolated from vaginas of healthy women[6] are capable of inhibiting HIV-1 infection. The choice of these *Lactobacillus* strains is based on the earlier reported anti-HIV-1 activity of these bacteria in human tissues ex vivo[12]. Moreover, these bacterial strains are the ones that mostly dominate the vaginal ecosystem[1]. We demonstrate that EVs released by lactobacilli into culture medium protect human T cells as well as human cervico-vaginal and tonsillar tissues ex vivo from HIV-1 infection. This protection is mediated, in part, by inhibition of viral attachment and entry to target cells due to diminished exposure of Env on EV-treated HIV-1 virions. Furthermore, using proteomic and metabolomic analysis, we identify several EV-associated bacterial proteins and metabolites that may play a role in this protective effect against HIV-1 infection.

## Results

**Lactobacilli EVs in HIV-1 inhibition.** Here, we characterized the EVs released by symbiotic vaginal lactobacilli in terms of their size and concentration, tested their anti-HIV-1 effect in human T cells and tissues, evaluated EV cytotoxicity, and identified EV-associated bioactive molecules that may contribute to the anti-HIV-1 inhibitory activity of the EVs studied.

**Characterization of EVs released by *Lactobacillus*.** We used NTA to characterize the sizes and concentrations of bacterial EVs produced by *L. crispatus* BC3, *L. crispatus* BC5, *L. gasseri* BC12, and *L. gasseri* BC13. We isolated EVs by ultracentrifugation from bacteria cultures (50 mL, $1 \times 10^9$ CFU per mL). All the tested bacteria released EVs of similar size, with mean diameters ranging from $133.14 \pm 2.90$ nm (*L. crispatus* BC3) to $141.26 \pm 9.78$ nm (*L. crispatus* BC5) (Fig. 1a, b). The concentration of EVs released varied from one bacterial strain to another: $3.26 \pm 0.11 \times 10^{10}$ (*L. crispatus* BC3), $1.18 \pm 0.32 \times 10^{10}$ (*L. crispatus* BC5), $5.87 \pm 0.20 \times 10^{10}$ (*L. gasseri* BC12), and $1.32 \pm 0.44 \times 10^{11}$ (*L. gasseri* BC13) particles per mL (Fig. 1c). Although MRS medium not conditioned by bacteria also contained particles, their concentrations were about two orders of magnitude lower ($4.13 \pm 0.70 \times 10^9$ particles per mL) than those of EVs released by bacteria (Fig. 1c). However, no specific protein bands were found in MRS-isolated particles in contrast to bacterial EVs. Moreover, we did not find eukaryotic EV markers (TSG101, CD63) in any *Lactobacillus*-derived EVs tested in our study compared with the positive control (MT-4 cell lysates), demonstrating that the preparation of bacteria-derived EVs did not contain eukaryotic EVs (Fig. 1d).

The average size of these particles was statistically not different from that of bacteria-derived EVs ($162 \pm 4.27$ nm) (Fig. 1a, b).

**Bacterial EVs suppress HIV-1 infection in human cell lines.** To investigate whether *Lactobacillus*-derived EVs affect HIV-1 infection, we infected two human CD4$^+$ T cell lines, MT-4 and Jurkat-tat, with this virus. We worked with a prototypic HIV-1 that uses CD4 and CCR5 cell-surface coreceptor to enter cells and is typical at the early stages of infection (isolate BaL) and a prototypic HIV-1 that uses CD4 and CXCR4 cell-surface coreceptor that often evolves at the late stages of infection (isolate LAI.04). Levels of HIV-1 infection were monitored from measurements of the release of HIV-1 gag protein p24 into the culture medium.

HIV-1$_{LAI.04}$ infection of MT-4 cell cultures varied from $3.75 \times 10^4$ to $1.29 \times 10^7$ p24$_{gag}$ pg per mL. As shown in Fig. 2a, HIV-1$_{LAI.04}$ replication in MT-4 cells was reduced by $91.88 \pm 3.15\%$ ($p < 0.0001$, $n = 14$, repeated-measures analysis of variance (ANOVA)) when cells were treated with *L. gasseri* BC12-derived EVs ($5 \times 10^8$ particles per mL) and by $59.35 \pm 2.34\%$ ($p < 0.0001$, $n = 5$, repeated-measures ANOVA) when MT-4 cells were treated with *L. crispatus* BC3-derived EVs. In contrast, no statistically significant HIV-1$_{LAI.04}$ inhibition was observed when MT-4 cell cultures were treated with similar numbers of EVs from *L. crispatus* BC5 ($p = 0.9967$, $n = 7$, repeated-measures ANOVA) or *L. gasseri* BC13-derived EVs ($p = 0.8750$, $n = 4$, repeated-measures ANOVA). Also, there was no inhibition of HIV-1 replication in MT-4 cells treated with particles isolated from MRS medium ($p = 0.9706$, $n = 4$, repeated-measures ANOVA) (Fig. 2a).

We chose EVs released from *L. gasseri* BC12, which showed a higher capacity to inhibit HIV-1 replication than the other bacterial-derived EVs, as a model condition to test the concentration-dependent anti-HIV-1 effect. Inhibition of HIV-1$_{LAI.04}$ infection of MT-4 cells by EVs was concentration dependent: *L. gasseri* BC12-derived EVs at a concentration of $5 \times 10^4$ EVs per mL suppressed HIV-1$_{LAI.04}$ replication by $14.59 \pm 3.23\%$ ($p = 0.04232$, $n = 15$, repeated-measures ANOVA), and the effect progressively augmented with increase in the concentration of bacterial EVs, reaching an inhibition of viral replication by $93.48 \pm 0.69\%$ ($p < 0.0001$, $n = 15$, repeated-measures ANOVA) at a concentration of $5 \times 10^8$ EVs per mL (Fig. 2b).

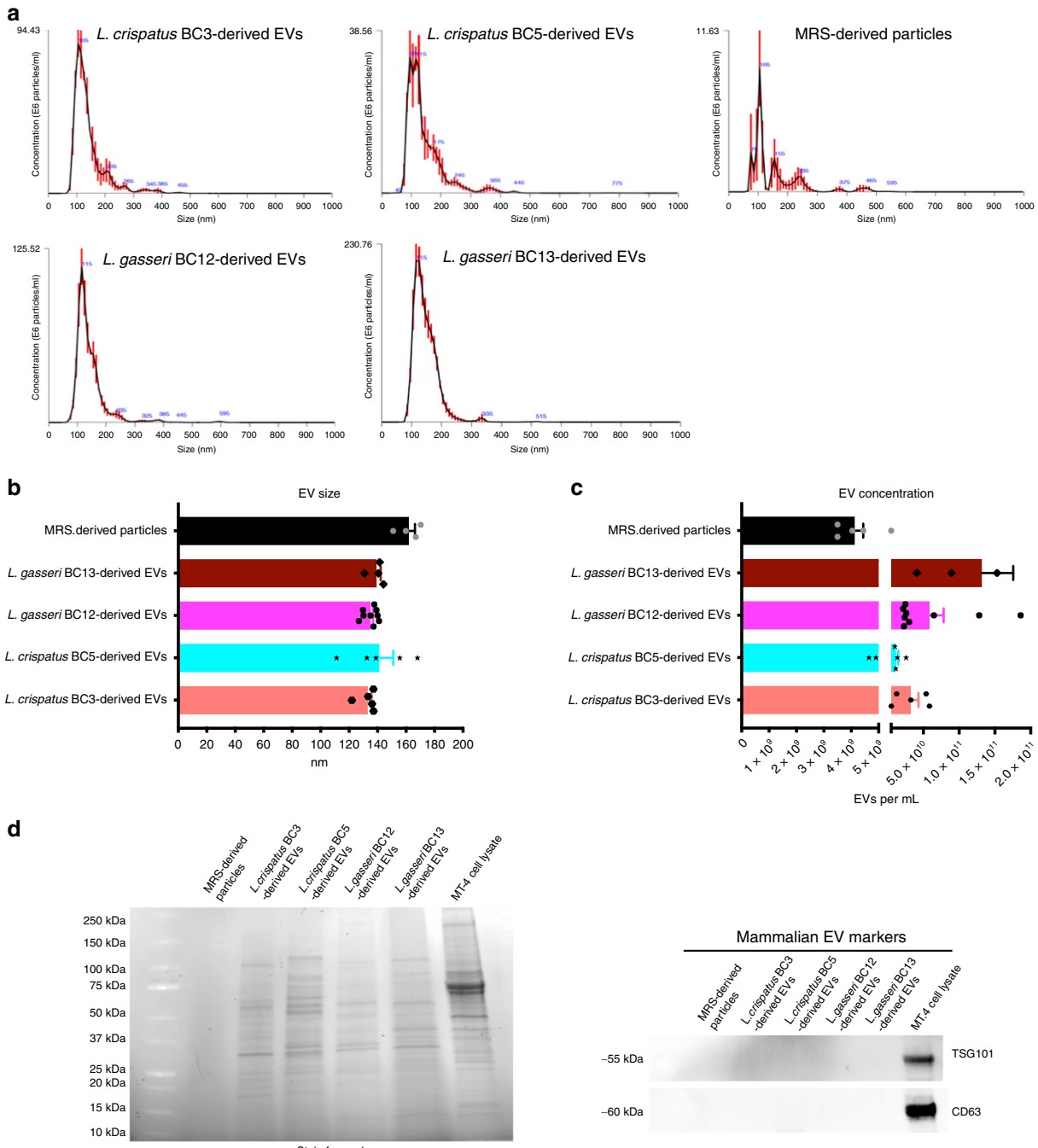

**Fig. 1 Nanoparticle tracking analysis of *Lactobacillus*-derived EVs.** Nanoparticle tracking analysis of EVs released by *L. crispatus* BC3, *L. crispatus* BC5, *L. gasseri* BC12, and *L. gasseri* BC13 and of particles present in MRS culture medium. **a** Representative analysis of EV size and concentration in EV samples diluted 1:100 with PBS. **b** Mean ± SEM of EV size (nm); **c** Mean ± SEM of EV concentration (particles per mL). Presented are the results of at least four independent measurements. **d** Proteins associated to *Lactobacillus*-derived EVs. Stain-free gel representing the spectrum of proteins obtained from EV lysates and absence of mammalian EV-associated proteins TSG101 and CD63. Source data are provided as a Source Data file.

The inhibition of HIV-1$_{BaL}$ infectivity in Jurkat-tat cells by *L. gasseri* BC12-derived EVs was also concentration dependent: EVs at a concentration of $5 \times 10^4$ EVs per mL inhibited HIV-1$_{BaL}$ replication by $33.18 \pm 9.16\%$ ($p = 0.0007$, $n = 8$, repeated-measures ANOVA), whereas EVs at $5 \times 10^8$ EVs per mL inhibited viral replication by $98.71 \pm 0.56\%$ ($p < 0.0001$, $n = 10$, repeated-measures ANOVA) (Fig. 2c). At the highest concentration of EVs, $5 \times 10^8$ EVs were added to $5 \times 10^5$ cells, and therefore the ratio was ~1000 EVs per cell.

Treatment of HIV-1-infected MT-4 cells with bacterial EVs did not affect the expression of p53 level, typically elevated after HIV-1 infections[26–28] (Supplementary Fig. 1).

**Bacterial EVs are not cytotoxic.** To check whether EV treatment decreases the viability of EV-treated cells, we evaluated the possible cytotoxic effect of EVs secreted by four *Lactobacillus* strains (*L. crispatus* BC3, *L. crispatus* BC5, *L. gasseri* BC12, *L. gasseri* BC13) on MT-4 and Jurkat-tat cell lines. Toward this goal,

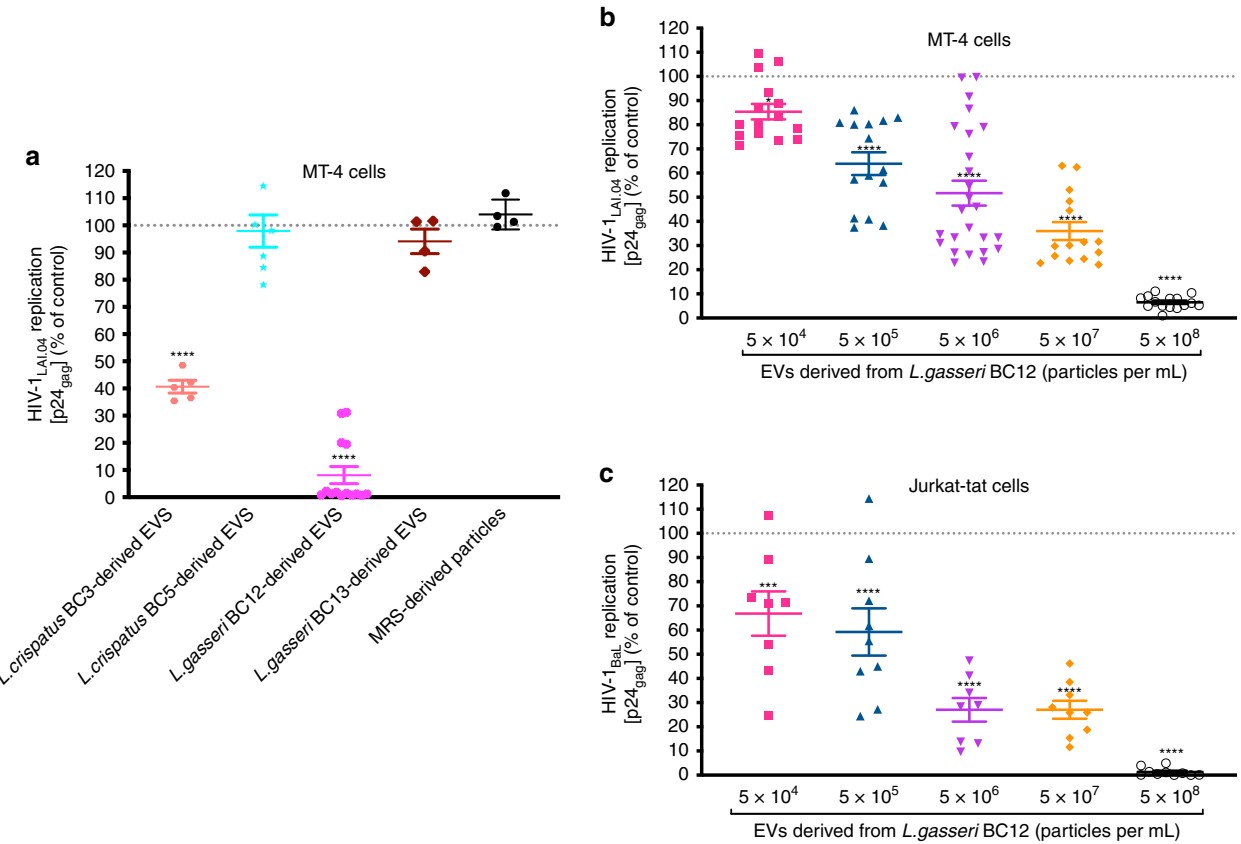

**Fig. 2 Antiviral activity of *Lactobacillus*-derived EVs in human T cells.** A mixture of HIV-1$_{LAI.04}$ or HIV-1$_{BaL}$ and *Lactobacillus*-derived EVs incubated for 1 h was added to MT-4 or Jurkat-tat cell cultures for 1 h. Cells were washed and cultured for 3 days in the presence of EVs. In control experiments, EV-free and particles isolated from fresh MRS medium were tested as well. Replication of HIV-1 was evaluated by measuring the capsid protein p24$_{gag}$, in the cell culture medium; data are presented as percentages of HIV-1 replication in untreated controls. **a** The effects of bacterial-derived EVs (5 × 10$^8$ particles per mL) from *L. crispatus* BC3, *L crispatus* BC5, *L. gasseri* BC12, and *L. gasseri* BC13, and of particles isolated from MRS medium (3.87 × 10$^8$ particles per mL), on HIV-1$_{LAI.04}$ replication in MT-4 cells. **b**, **c** Concentration dependence of EVs derived from *L. gasseri* BC12 on replication of HIV-1$_{LAI.04}$ in MT-4 (**b**) and of HIV-1$_{BaL}$ in Jurkat-tat (**c**) cell lines. Presented are means ± SEM from at least four independent measurements. Asterisks indicate statistical significance by one-way ANOVA multiple comparison with Dunnett's correction (*$p < 0.05$, ***$p < 0.001$, ****$p < 0.0001$). Source data are provided as a Source Data file.

human T cells were cultured in a medium containing or not containing 5 × 10$^8$ bacterial EVs per mL. We also tested the particles isolated from bacteria-free MRS medium (5 × 10$^8$ particles per mL) in an experimental control condition. We used Triton X-100 at 0.2% as a positive control in the toxicity assays. After 3 days of cell treatment, we evaluated cell viability from counts of the total and dead cells in control and EV-treated cultures using a propidium-iodide-based assay. There was no decrease in the number of viable cells after treatment with different amounts of *Lactobacillus*-derived EVs or similarly isolated particles from MRS medium in comparison with untreated controls, both in MT-4 and in Jurkat-tat cell cultures (Fig. 3a). The viability of cells treated with the highest amount of bacterial EVs (5 × 10$^8$ EVs per mL) was statistically not different ($p > 0.13$, repeated-measures ANOVA) from that in untreated controls (100%), ranging from 88.90 ± 2.54% (*L. crispatus* BC13-derived EVs) to 113.20 ± 6.57% (*L. crispatus* BC3-derived EVs) in MT-4 cells and from 90.95 ± 1.68% (*L. crispatus* BC13-derived EVs) to 102.80 ± 3.52% (*L. crispatus* BC3-derived EVs) in Jurkat-tat cells. Similarly, the cell viability upon treatment with MRS medium-derived particles was not altered compared with the control ($p > 0.39$, repeated-measures ANOVA) in both cell lines (89.91 ± 4.70% and 90.86 ± 3.85% in MT-4 and Jurkat-tat, respectively). On the other hand, as expected, Triton X-100 at 0.2% reduced the cell viability significantly, down to 7.81 ± 0.53%

($p < 0.0001$, repeated-measures ANOVA) in MT-4 cells and 8.66 ± 0.30% ($p < 0.0001$, repeated-measures ANOVA) in Jurkat-tat cells.

Similar results were obtained with the MTT assay (Fig. 3b). Indeed, the percentages of viable cells were not altered significantly ($p > 0.17$, repeated-measures ANOVA) by particles from MRS medium or by bacterial EVs compared with the control. The cell viability in MT-4 cells ranged from 99.20 ± 3.59% (*L. crispatus* BC3-derived EVs) to 116.70 ± 2.56% (MRS-derived particles) for MT-4 cells and from 97.16 ± 1.90% (*L. crispatus* BC5-derived EVs) to 105.00 ± 1.83% (*L. crispatus* BC13-derived EVs) for Jurkat-tat cells. Instead, treatment with Triton X-100 at 0.2% reduced the cell viability significantly, down to 27.91 ± 2.61% ($p < 0.0001$, repeated-measures ANOVA) in MT-4 cells and by 15.46 ± 0.31% ($p < 0.0001$, repeated-measures ANOVA) in Jurkat-tat cells.

The lack of EV cytotoxicity was confirmed by flow cytometry. EV-treated and control cell cultures were stained with live/dead Fixable Viability Dye, anti-CD3-, and anti-CD4-antibodies, and were then enumerated with a flow cytometer. There was no significant difference between fractions of live, CD3$^+$, and CD4$^+$ MT-4 cells 3 days post *L. gasseri* BC12-derived EV treatment (5 × 10$^8$ EVs per mL; CD3$^+$ cells: 99.91%, CD4$^+$ cells: 98.54%) and respective control values (CD3$^+$ cells: 99.80%, CD4$^+$ cells: 99.54%) (Fig. 3c, Supplementary Fig. 2).

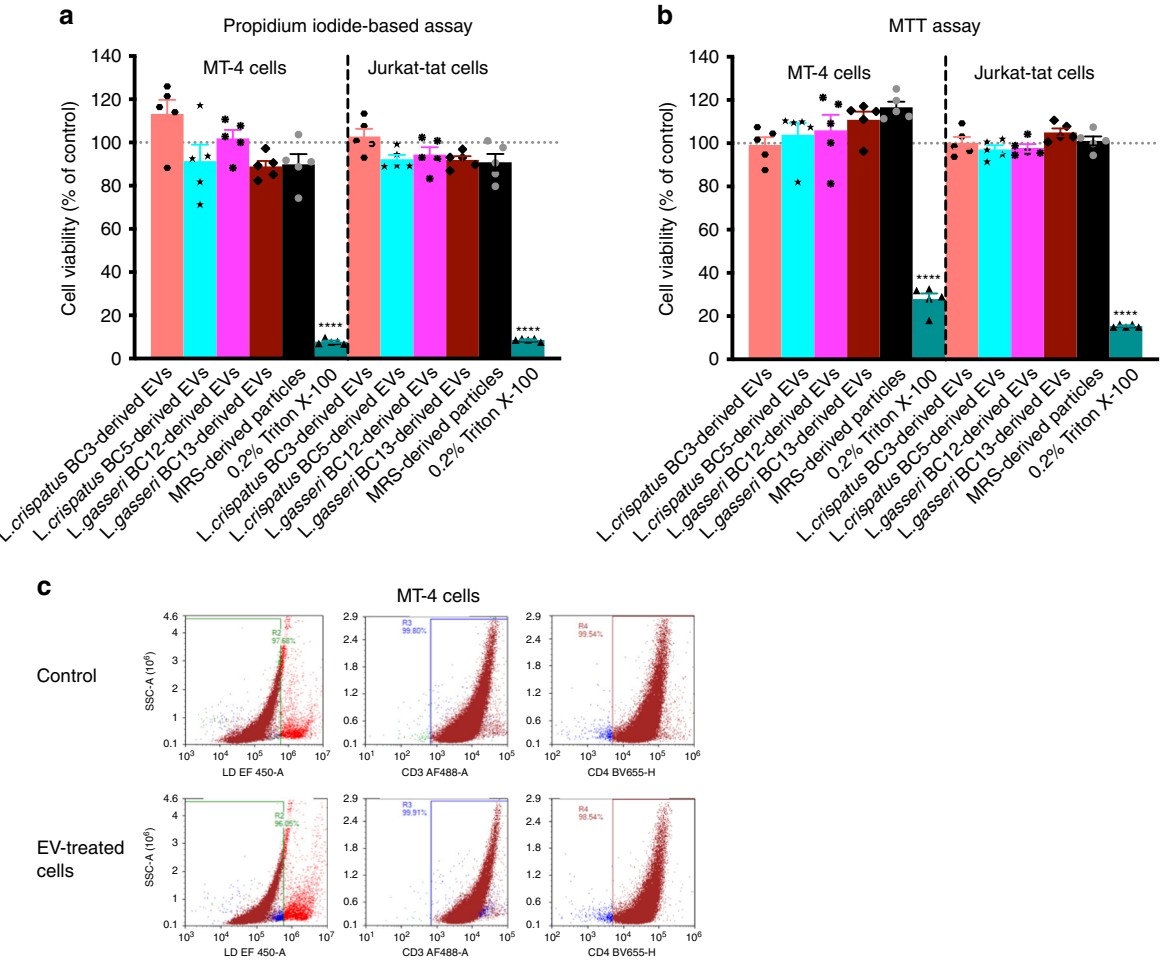

**Fig. 3 Viability of cells treated with *Lactobacillus*-derived EVs.** MT-4 and Jurkat-tat cells were treated or not treated for 3 days with EVs ($5 \times 10^8$ EVs per mL) isolated from four *Lactobacillus* strains (*L. crispatus* BC3, *L. crispatus* BC5, *L. gasseri* BC12, *L. gasseri* BC13), and with particles derived from MRS medium. Triton X-100 (0.2%) was used as a cytotoxic agent. **a** The numbers of viable cells were counted according to a propidium-iodide-based assay. Results are expressed as percentages of viable cells in EV-free or EV-treated cells. Presented are means ± SEM from five independent measurements. **b** Cell viability according to MTT assay. Results are expressed as percentages of viable cells in EV-free or EV-treated cells. Presented are means ± SEM from five independent measurements. **c** Cell depletion in MT-4 cell cultures treated with *L. gasseri* BC12-derived EVs ($5 \times 10^8$ EVs per mL), as measured with flow cytometry. Panels from left to right represent staining for live/dead cells, CD3, and CD4 populations in EV-untreated cells (upper row) and in cells treated with *Lactobacillus*-derived EVs (lower row). Asterisks indicate statistical significance by one-way ANOVA multiple comparison with Dunnett's correction (****$p < 0.0001$). Source data are provided as a Source Data file.

**Bacterial EVs suppress HIV-1 infection in human tissues**. We investigated whether *Lactobacillus*-derived EVs can suppress HIV-1 infection in human tonsillar and cervico-vaginal tissues ex vivo. Human tonsillar tissues ex vivo infected with HIV-1$_{BaL}$ were cultured in medium containing or not containing equal amounts of EVs ($5 \times 10^8$ EVs per mL) isolated from culture media derived from four *Lactobacillus* strains: *L. crispatus* BC3, *L. crispatus* BC5, *L. gasseri* BC12, *L. gasseri* BC13, and MRS medium-derived particles. As shown in Fig. 4a, HIV-1$_{BaL}$ replication was significantly reduced, by $38.70 \pm 11.36\%$ ($p = 0.0199$, $n = 5$, repeated-measures ANOVA), when tonsillar tissue blocks were cultured in presence of EVs derived from *L. crispatus* BC3, and by $48.19 \pm 10.11\%$ ($p = 0.0033$, $n = 5$, repeated-measures ANOVA) when cultured in presence of EVs derived from *L. gasseri* BC12, compared with tonsillar tissue blocks cultured in the absence of bacterial EVs.

There was no statistically significant inhibition of HIV-1$_{BaL}$ replication in tonsillar tissues cultured ex vivo in the presence of EVs derived from *L. crispatus* BC5 ($p = 0.4602$, $n = 5$, repeated-measures ANOVA) or *L. gasseri* BC13 ($p = 0.4744$, $n = 5$), or

particles derived from MRS medium ($p > 0.9999$, $n = 4$, repeated-measures ANOVA).

In ex vivo tissues, we also evaluated the concentration dependency of HIV-1 suppression by bacterial EVs. We cultured tonsillar tissue blocks infected with HIV-1$_{LAI.04}$ and cervico-vaginal tissue blocks infected with HIV-1$_{BaL}$ in the presence of tenfold serial dilutions of EVs derived from *L. gasseri* BC12 ($5 \times 10^5$; $5 \times 10^6$, $5 \times 10^7$, $5 \times 10^8$ EVs per mL). The inhibitory effect of *L. gasseri* BC12-derived EVs on HIV-1$_{LAI.04}$ replication in tonsillar tissue blocks was concentration dependent, with a $20.92 \pm 7.54\%$ inhibition ($p = 0.0236$, $n = 5$, repeated-measures ANOVA) with EVs at a concentration of $5 \times 10^6$ EVs per mL and a $61.03 \pm 5.58\%$ ($p < 0.0001$, $n = 5$, repeated-measures ANOVA) inhibition at the maximal tested EV concentration ($5 \times 10^8$ EVs per mL) (Fig. 4b). A similar concentration dependency of bacterial EVs on HIV-1 replication was obtained in cervico-vaginal tissues, where HIV-1$_{BaL}$ replication was inhibited by $31.30 \pm 3.22\%$ ($p = 0.028$, $n = 4$, repeated-measures ANOVA) when EVs were present at $5 \times 10^5$ EVs per mL and by $59.83 \pm 4.92\%$ ($p < 0.0001$, $n = 4$, repeated-measures ANOVA) when EVs

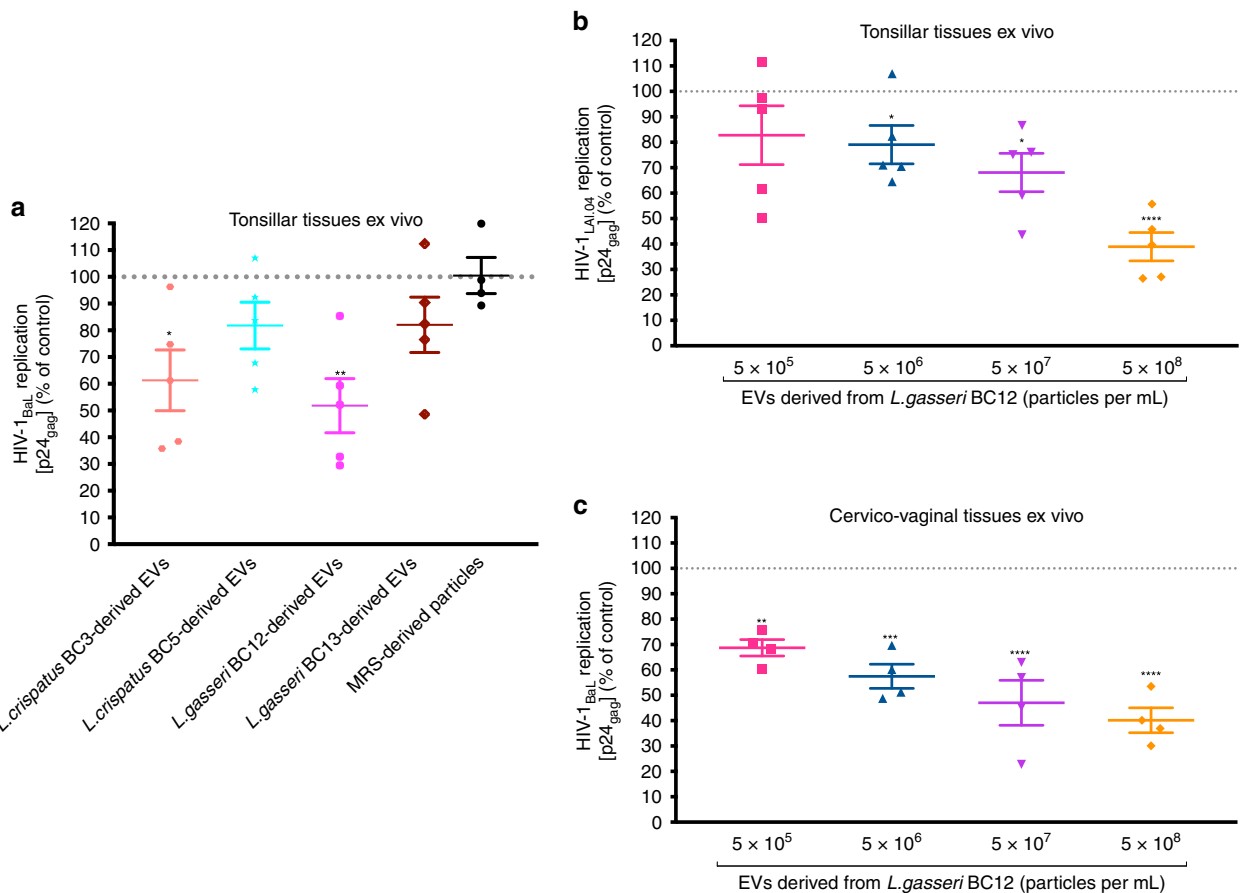

**Fig. 4 Antiviral activity of *Lactobacillus*-derived EVs in human ex vivo tissues.** Tonsillar tissue blocks were infected with EV-pretreated HIV-1$_{BaL}$ and cultured for 12 days, with replacement every 3 days of tissue culture medium containing EVs obtained from *L. crispatus* BC3, *L crispatus* BC5, *L. gasseri* BC12, *L. gasseri* BC13, or particles derived from MRS medium. Replication of HIV-1 was evaluated from measurements of the capsid protein p24$_{gag}$ in tissue culture medium and is represented as a percentage of HIV-1 replication in untreated control. **a** Anti-HIV-1 effects of EVs released by *L. crispatus* BC3, *L crispatus* BC5, *L. gasseri* BC12, and *L. gasseri* BC13 (5 × 10⁸ EVs per mL). **b, c** Concentration dependence of antiviral activity of *L. gasseri* BC12-derived EVs in human tonsillar tissues ex vivo infected with HIV-1$_{LAI.04}$ (**b**) and in cervico-vaginal tissues ex vivo infected with HIV-1$_{BaL}$ (**c**). Presented are means ± SEM from tissues of at least four donors. Asterisks indicate statistical significance by one-way ANOVA multiple comparison with Dunnett's correction (*$p < 0.05$, **$p < 0.01$, ***$p < 0.001$, ****$p < 0.0001$). Source data are provided as a Source Data file.

were present at $5 \times 10^8$ EVs per mL (Fig. 4c). In the above-described experiments with tonsillar tissue explants, nine tissue blocks were cultured in 3 mL of medium. Therefore, $1.5 \times 10^9$ EVs were used ($5 \times 10^8$ EVs per mL × 3) for a total of ~$1.08 \times 10^6$ CD4 + T cells (~$1.2 \times 10^5$ CD4 + T cells per block × 9 blocks)[25], thus giving a ratio of 1000 EVs per cell.

**Bacterial EVs reduce HIV-1 entry/attachment to target cells.** To investigate whether the inhibition of HIV-1 replication was associated with a decrease in viral entry, we used TZM-bl cells. HIV-1 infection of these cells was reduced in a concentration-dependent manner. Luciferase expression was decreased by 38.52 ± 8.74% ($p = 0.001$, $n = 6$, repeated-measures ANOVA) when HIV-1$_{LAI.04}$-infected TZM-bl cells were incubated with EVs from *L. gasseri* BC12 at a concentration of $5 \times 10^6$ EVs per mL and by 69.45 ± 6.10% ($p < 0.0001$, $n = 6$, repeated-measures ANOVA) at a concentration of $5 \times 10^8$ EVs per mL. In contrast, no statistically significant HIV-1$_{LAI.04}$ entry inhibition was observed when TZM-bl cell cultures were treated with particles from MRS ($p = 0.9661$, $n = 3$, repeated-measures ANOVA) at $5 \times 10^8$ EVs per mL (Fig. 5a).

The effect of *Lactobacillus* EVs on HIV-1 attachment/entry was also studied in MT-4 cell line. MT-4 cells were incubated with HIV-1$_{LAI.04}$ for 2 h at 4 °C, in the presence or absence of EVs

derived from *L. crispatus* BC3, *L. crispatus* BC5, *L. gasseri* BC12, or *L. gasseri* BC13, or of particles derived from MRS medium. After eliminating all free viral particles with several washes, we observed that the levels of HIV-1 p24$_{gag}$, were reduced significantly upon the incubation with EVs derived from *L. crispatus* BC3, by 63.31 ± 4.74% ($p < 0.0001$, $n = 3$, repeated-measures ANOVA), and with EVs derived from *L. gasseri* BC12, by 80.41 ± 5.35% ($p < 0.0001$, $n = 6$, repeated-measures ANOVA) (Fig. 5b). In contrast, there was no statistically significant reduction of p24$_{gag}$ levels when virions were treated with EVs derived from *L. crispatus* BC5 ($p = 0.0517$, $n = 3$, repeated-measures ANOVA), with *L. gasseri* BC13-derived EVs ($p = 0.9276$, $n = 3$), or with particles derived from MRS medium ($p = 0.9999$, $n = 3$, repeated-measures ANOVA).

**Bacterial EVs prevent HIV-1 infection affecting viral Env.** To investigate whether *Lactobacillus*-derived EVs affect HIV-1 envelope directly, we pretreated HIV-1$_{LAI.04}$ virions with EVs derived from four *Lactobacillus* strains (*L. crispatus* BC3, *L. crispatus* BC5, *L. gasseri* BC12, and *L. gasseri* BC13) for 1 h. Then, we captured HIV-1$_{LAI.04}$ virions using PG9 antibodies coupled to magnetic nanoparticles (MNPs). PG9 preferentially recognizes the HIV-1 trimeric envelope proteins (gp120). As shown in Fig. 6, the amounts of HIV-1$_{LAI.04}$ viral particles

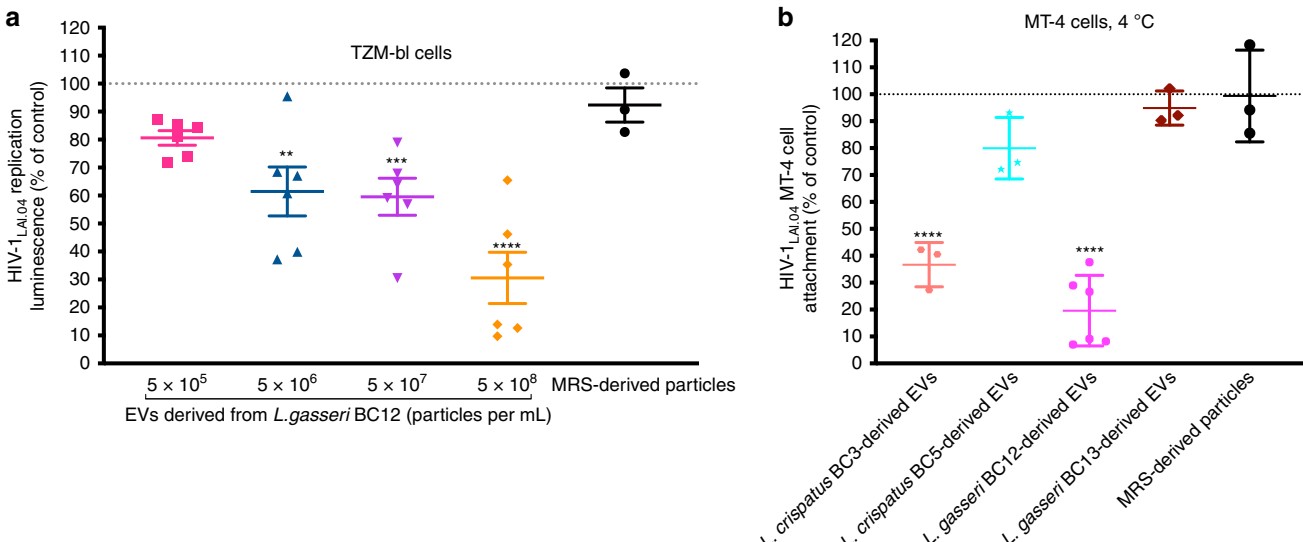

**Fig. 5 Effect of *Lactobacillus*-derived EVs on HIV-1 entry/attachment to the target cell. a** TZM-bl cells were cultured in an EV–HIV-1 mixture overnight and then continued to be cultured in the presence of EVs from *L. gasseri* BC12 at different concentrations or particles derived from MRS medium for 3 days. To measure the HIV-1$_{LAI.04}$ integration, we removed the cell culture medium, lysed the cells, added substrate, and measured the luminescence. **b** MT-4 cells were treated for 2 h with a mixture of HIV-1$_{LAI.04}$ and EVs derived from four *Lactobacillus* strains (*L. crispatus* BC3, *L. crispatus* BC5, *L. gasseri* BC12, *L. gasseri* BC13), or with particles derived from MRS medium, and after several thorough washings cells were lysed. Viral p24$_{gag}$ protein in cell lysates was measured with Luminex. Presented are means ± SEM from at least three independent measurements. Asterisks indicate statistical significance by one-way ANOVA multiple comparison with Dunnett's correction (**$p < 0.01$, ***$p < 0.001$, ****$p < 0.0001$). Source data are provided as a Source Data file.

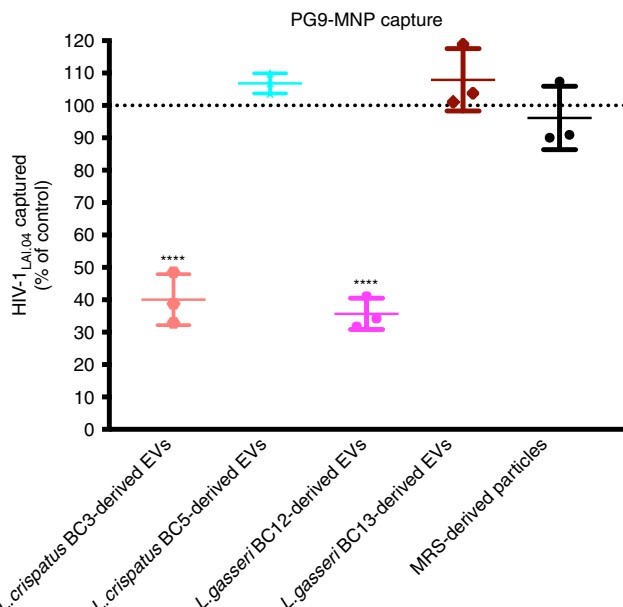

**Fig. 6 Effect of *Lactobacillus*-derived EVs on HIV-1 viral envelope.**
HIV-1$_{LAI.04}$ virions were pretreated with EVs derived from each of four *Lactobacillus* strains (*L. crispatus* BC3, *L. crispatus* BC5, *L. gasseri* BC12, *L. gasseri* BC13) or particles derived from MRS medium, or PBS (control) for 1 h. Next, HIV-1$_{LAI.04}$ virions were captured with PG9 antibody coupled to magnetic nanoparticles. This antibody recognizes HIV-1 trimeric envelope proteins. Then, we separated the captured HIV-1 virions using magnetic columns and measured the concentrations of HIV-1 p24$_{gag}$ antigen using an immunofluorescent cytometric bead-based assay (Luminex). Presented are means ± SEM from three independent measurements. Asterisks indicate statistical significance by one-way ANOVA multiple comparison with Dunnett's correction (****$p < 0.0001$). Source data are provided as a Source Data file.

captured, as measured from HIV-1 p24$_{gag}$ concentration, were reduced significantly, by $59.93 \pm 4.55\%$ ($p < 0.0001$, $n = 3$, repeated-measures ANOVA) upon viral pretreatment with EVs derived from *L. crispatus* BC3 and by $64.32 \pm 2.78\%$ ($p < 0.0001$, $n = 3$, repeated-measures ANOVA) when HIV-1$_{LAI.04}$ were pretreated with EVs derived from *L. gasseri* BC12, compared with the captured viral particles in the absence of bacterial EVs.

There was no statistically significant effect on the HIV-1$_{LAI.04}$ envelope when virions were pretreated with EVs derived from *L. crispatus* BC5 ($p = 0.6590$, $n = 3$, repeated-measures ANOVA) or *L. gasseri* BC13-derived EVs ($p = 0.5654$, $n = 3$, repeated-measures ANOVA), or with particles derived from MRS medium ($p = 0.9011$, $n = 3$, repeated-measures ANOVA).

**EV treatment of cells does not prevent HIV-1 infection.** To study whether bacterial EVs alter host cell function to induce protection against HIV-1, TZM-bl, or MT-4 cells were cultured in cell culture medium containing or not containing bacterial EVs for 24 h, prior to HIV-1 infection, and HIV-1 infection was evaluated. As shown in Fig. 7a, HIV-1$_{LAI.04}$ replication was not significantly reduced upon pretreatment with bacterial EVs derived from *L. crispatus* BC3 ($p = 0.3665$, $n = 4$, repeated-measures ANOVA), *L. crispatus* BC5 ($p = 0.0972$, $n = 4$, repeated-measures ANOVA), *L. gasseri* BC12 ($p = 0.2031$, $n = 4$, repeated-measures ANOVA), *L. gasseri* BC13 ($p = 0.0594$, $n = 4$, repeated-measures ANOVA) or upon pretreatment with MRS-derived particles ($p = 0.0515$, $n = 4$, repeated-measures ANOVA), compared with control experiments.

Similar results were obtained in MT-4 cells. HIV-1$_{LAI.04}$ replication was not significantly reduced upon pretreatment with bacterial EVs derived from *L. crispatus* BC3 ($p = 0.31901$, $n = 3$, repeated-measures ANOVA), *L. crispatus* BC5 ($p = 0.1286$, $n = 3$, repeated-measures ANOVA), *L. gasseri* BC12 ($p = 0.8680$, $n = 3$, repeated-measures ANOVA), *L. gasseri* BC13 ($p = 0.1127$, $n = 3$, repeated-measures ANOVA) or upon pretreatment with

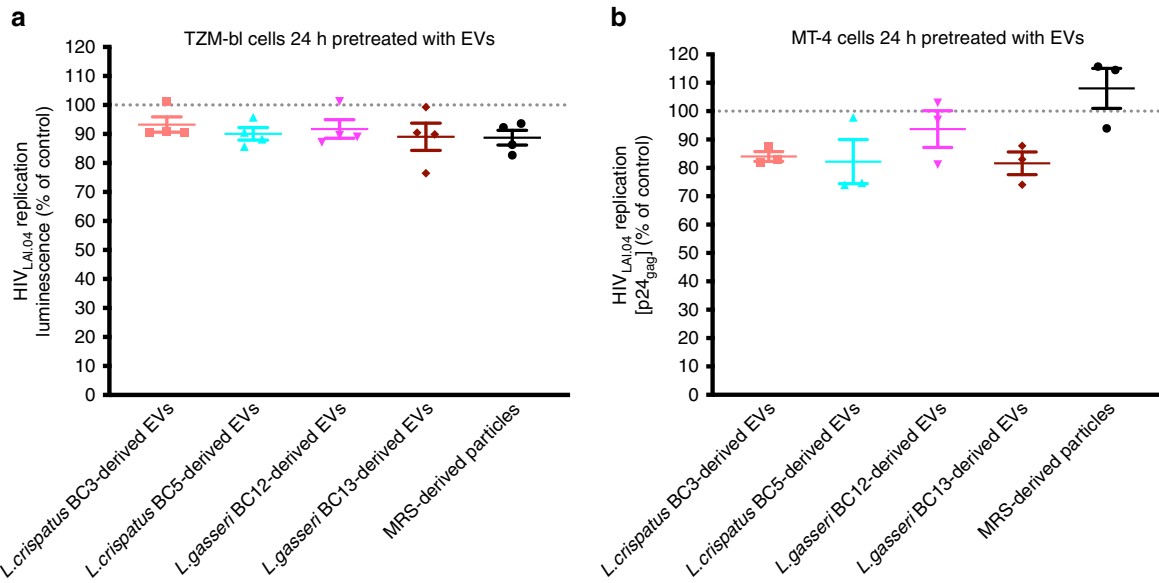

**Fig. 7 HIV-1 infection in cells pretreated with *Lactobacillus*-derived EVs.** To study whether bacterial EVs alter host cell function to induce protection against HIV-1, TZM-bl or MT-4 cells were treated with bacterial EVs derived from *L. crispatus* BC3, *L. crispatus* BC5, *L. gasseri* BC12, *L. gasseri* BC13, or MRS-isolated particles for 24 h, prior to HIV-1 infection, and then HIV-1 infectivity was evaluated. **a** To measure HIV-1$_{LAI.04}$ integration in TZM-bl cells, we removed the cell culture medium, lysed the cells, added substrate, and measured the luminescence. **b** To measure HIV-1$_{LAI.04}$ replication in MT-4 cells the levels of HIV-1 p24$_{gag}$ protein in cell culture were measured with Luminex. Presented are means ± SEM from at least three independent measurements. Source data are provided as a Source Data file.

MRS-derived particles ($p = 0.7353$, $n = 3$, repeated-measures ANOVA) (Fig. 7b).

**Metabolites associated with *Lactobacillus*-derived EVs.** We used $^1$H-NMR to identify metabolites that are associated with bacterial EVs. The metabolomic analysis was conducted on EVs derived from *Lactobacillus* strains that inhibited HIV-1 replication (*L. crispatus* BC3, *L. gasseri* BC12), on EVs that did not inhibit HIV-1 replication (*L. crispatus* BC5, *L. gasseri* BC13), and as well on particles isolated from MRS medium. To release all the EV content, we sonicated them prior to the analysis. We identified 42 molecules, mainly belonging to the classes of organic acids, amino acids, sugars, and nitrogen bases (Supplementary Table 1). Five of them were found to correlate significantly with the antiviral activity. To highlight their underlying trends, these molecules were employed as a basis for an rPCA model (Fig. 8a). Two principal components (PC) appeared to describe the variance of the samples optimally. PC 1, accounting for 74.8% of the variance described by the model, accounted for the sample's antiviral activity against HIV-1. EVs derived from *L. crispatus* BC3 and *L. gasseri* BC12, which efficiently suppress HIV-1 inhibition, showed lower PC 1 scores. EVs from *L. crispatus* BC5 and *L. gasseri* BC13, which did not inhibit HIV-1, showed higher PC 1 scores. *L. crispatus* BC3- and *L. gasseri* BC12-derived EVs were mainly associated with high amounts of methionine, glycine, hypoxanthine, and glutamate. In contrast, EVs derived from *L. crispatus* BC5 and *L. gasseri* BC13, as well as particles derived from MRS medium, were mainly characterized by the presence of asparagine (Fig. 8b).

The samples from the same bacterial strains showed similar PC scores, thus suggesting a good biological reproducibility of the metabolome's profile.

**Proteins associated with *Lactobacillus*-derived EVs.** We then investigated which proteins of *Lactobacillus*-associated EVs could be responsible for the above-described HIV-1 suppression by

*L. gasseri* BC12 in comparison with EVs released by *L. crispatus* BC5, which do not suppress HIV-1 infection.

Using liquid chromatography–mass spectroscopy, we investigated the spectra of bacterial proteins present in the EVs of *L. gasseri* BC12 and *L. crispatus* BC5. Altogether, we identified 18 bacterial proteins associated with EVs derived from these two *Lactobacillus* strains (Table 1, Supplementary Table 2). Fifteen proteins were identified in EVs derived from *L. gasseri* BC12 and eleven proteins were identified in EVs released by *L. crispatus* BC5. Seven proteins were found in both *L. gasseri* BC12- and *L. crispatus* BC5-derived EVs. These were ATP synthase subunit beta from *L. gasseri*, ATP synthase subunit alpha, phosphonates import ATP-binding protein PhnC, ATP synthase subunit b, enolase 1, ATP synthase subunit beta, and 30S ribosomal protein S4. Eight proteins (enolase 2, 60 kDa chaperonin, elongation factor Tu, ATP synthase gamma chain, foldase protein PrsA 1, ATP synthase subunit delta, pyruvate kinase, and triosephosphate isomerase) were identified only in EVs derived from *L. gasseri* BC12, and three proteins (50S ribosomal protein L4, 50S ribosomal protein L21, and 50S ribosomal protein L2) were identified only in EVs derived from *L. crispatus* BC5.

According to the protein localization in the cellular component by Gene Ontology (GO) terms (Fig. 9a, Supplementary Table 2), the majority of the identified proteins associated either with *L. gasseri* BC12- or *L. crispatus* BC5-derived EVs were in the cytoplasm (44.44%, 45.45%) and membrane (44.44%, 45.45%), respectively. The remainder of the proteins exert their activities extracellularly (11.11%, 9.09%).

According to the molecular functions by the GO terms (Fig. 9b, Supplementary Table 2), most of the EV-related proteins identified in both *L. gasseri* BC12- and *L. crispatus* BC5-derived EVs are involved in ATP binding (20.00%, 20.00%), proton-transporting ATP synthase activity (15.00%, 20.00%), rotational mechanism (15.00%, 20.00%), magnesium ion binding (10.00%, 5.00%), phosphopyruvate hydratase activity (5.00%, 5.00%), rRNA binding (2.50%, 20.00%), and ATPase-coupled organic phosphonate transmembrane transporter activity (2.50%, 5.00%),

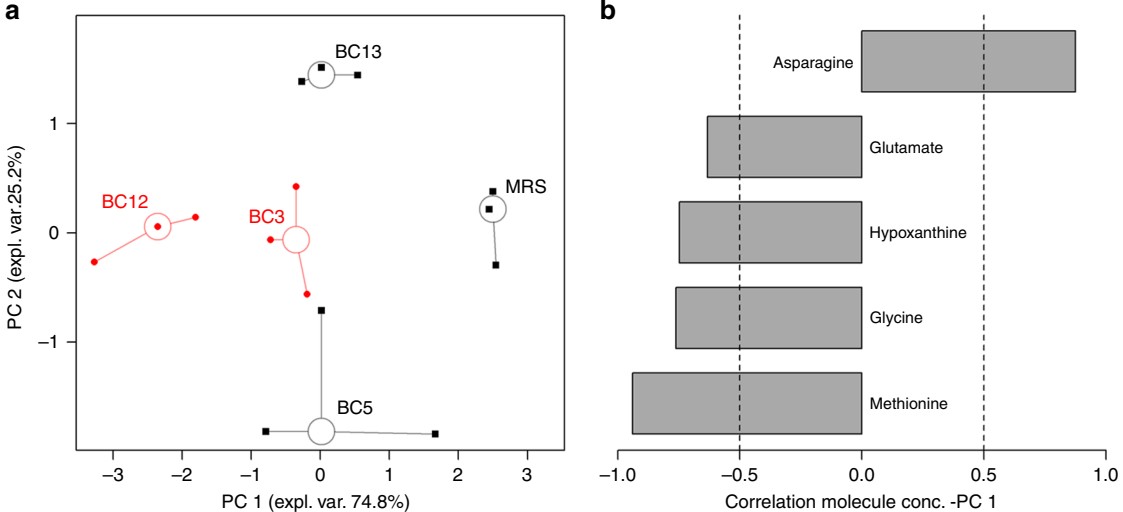

**Fig. 8 Principal component analysis of metabolomes in *Lactobacillus*-derived EVs. a** rPCA model calculated on the space constituted by the concentration of the molecules significantly related to antiviral activity against HIV-1. Empty circles highlight the median values of the repetitions for each sample, while black and red colors highlight samples with low and high anti-HIV-1 activity, respectively. **b** Bar plot describing the correlation between the concentration of each molecule and its importance along PC 1.

**Table 1 Protein cargo of EVs derived from *L. gasseri* BC12 and *L. crispatus* BC5.**

| *Lactobacillus*-derived EV-associated proteins | Accession number |
|---|---|
| 1. ATP synthase subunit beta | ATPB_LACGA |
| 2. ATP synthase subunit alpha | ATPA_LACGA (+1) |
| 3. Phosphonates import ATP-binding protein PhnC | PHNC_LACGA |
| 4. ATP synthase subunit b | ATPF_LACGA |
| 5. Enolase 2[a] | ENO2_LACGA |
| 6. 60 kDa chaperonin[a] | CH60_LACGA (+1) |
| 7. Enolase 1 | ENO1_LACGA (+1) |
| 8. Elongation factor Tu[a] | EFTU_LACGA |
| 9. ATP synthase gamma chain[a] | ATPG_LACGA |
| 10. Foldase protein PrsA 1[a] | PRSA1_LACJO |
| 11. ATP synthase subunit beta | ATPB_LACAC |
| 12. ATP synthase subunit delta[a] | ATPD_LACGA |
| 13. 50S ribosomal protein L4[b] | RL4_LACAC |
| 14. Pyruvate kinase[a] | KPYK_LACDE |
| 15. 30S ribosomal protein S4 | RS4_LACGA (+1) |
| 16. Triosephosphate isomerase[a] | TPIS_LACGA (+1) |
| 17. 50S ribosomal protein L21[b] | RL21_LACAC (+1) |
| 18. 50S ribosomal protein L2[b] | RL2_LACGA (+2) |

[a]Protein identified only in *L. gasseri* BC12-derived EVs
[b]Protein identified only in *L. crispatus* BC5-derived EVs

respectively. The proteins associated with EVs derived from *L. gasseri* BC12 are also involved in kinase activity (5.00%), potassium ion binding (5.00%), pyruvate kinase activity (5.00%), unfolded protein binding (2.50%), GTP binding (2.50%), GTPase activity (2.50%), translation elongation factor activity (2.50%), peptidyl-prolyl *cis–trans* isomerase activity (2.50%), and triose-phosphate isomerase activity (2.50%). The proteins identified in EVs from *L. crispatus* BC5 only, are also involved in transferase activity (5.00%).

## Discussion

Mammalian mucosae are populated by numerous bacteria that collectively form the microbiota that controls many of the host's normal and pathological processes. In particular, a vaginal microbiota dominated by *Lactobacillus* species is protective against various pathogens, including HIV-1[1,2,13].

Microbiota–host-cell crosstalk is a complex multifactorial phenomenon. A growing body of evidence suggests that one of the important mediators of cell–cell communications is nanosized extracellular vesicles (EVs), produced by all domains of life, including archaea, fungi, parasites, and bacteria[14,29,30]. Initially, it was thought that, in contrast to Gram-negative bacteria, Gram-positive bacteria including *Lactobacillus* could not produce EVs because of their thick cell walls. However, it was shown recently that Gram-positive bacteria are capable of releasing phospholipid bilayer nanosized EVs with a diameter range from 20 to 200 nm, like Gram-negative- or eukaryotic-derived EVs[15,17,20].

Here, we report for the first time on the production of EVs by *Lactobacillus* strains of human vaginal origin. We demonstrated that vaginal lactobacilli (*L. crispatus*, *L. gasseri*) release nanosized EVs similar to those released by *Lactobacillus* strains of gastro-intestinal origin such as *L. casei*, *L. rhamnosus*, *L. reuteri*, and *L. plantarum*[18,19,21–24]. Preparation of EVs isolated from lactobacilli cultured in MRS medium did not contain eukaryotic EV markers. Also, no specific protein bands were found in MRS-isolated particles in contrast with bacterial EVs. This may be expected since the MRS medium was autoclaved, which most likely resulted in denaturation of the proteins present in the MRS medium. Nevertheless we included these particles as experimental control conditions.

We investigated whether EVs released by lactobacilli contribute to the protective effect of *Lactobacillus* against vaginal HIV-1 infection[2,9–11]. We addressed this question by using immortalized human T cells as well as human cervico-vaginal tissue and lymphoid tonsillar tissues ex vivo. The ex vivo model is the standard tissue culture for study of HIV-1 pathogenesis, as it faithfully reflects many aspects of tissues where critical events in HIV-1 transmission and pathogenesis occur in vivo. Tissues ex vivo support productive HIV-1 infection without exogenous activation or stimulation and retain the pattern of expression of key cell-surface molecules relevant to HIV-1 infection[25]. Also, in a previous study, we demonstrated that various strains of *Lactobacillus* isolated from vaginas of healthy individuals inhibit HIV-1 replication in human cervico-vaginal and tonsillar tissues ex vivo[12]. Specifically, we found that *L. crispatus* BC3, *L. crispatus* BC5,

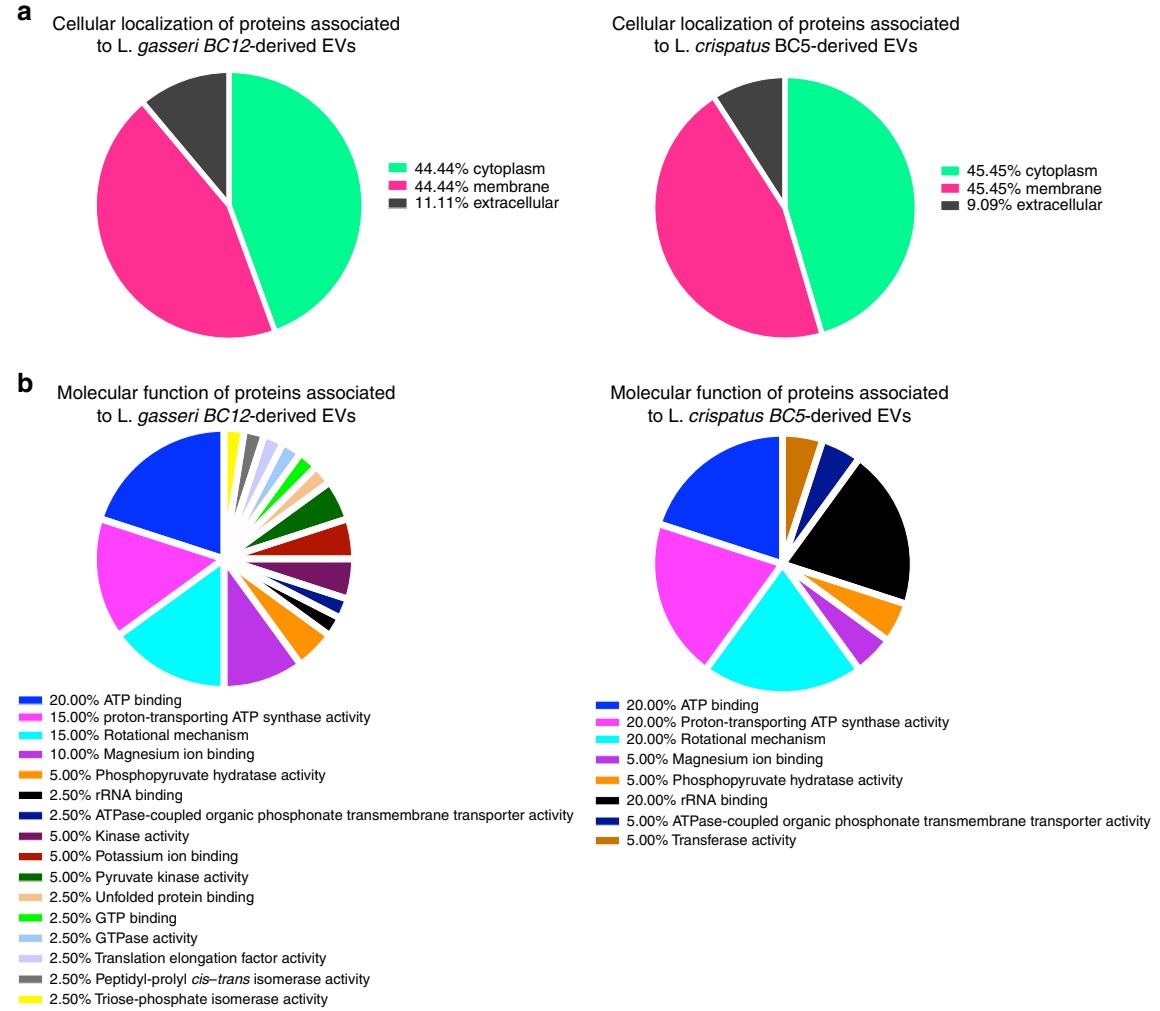

**Fig. 9 GO terms of EV-related proteins. a** According to cellular localization. **b** According to biological function.

L. gasseri BC12, and L. gasseri BC13 largely protected human ex vivo tissues from HIV-1 infection.

Here, we showed that this protection can be mediated by bacterial EVs. EVs released by L. crispatus BC3 and L. gasseri BC12 largely protected human tissues ex vivo and T cells from HIV-1 infection. The HIV-1 inhibitory effects of EVs from L. crispatus BC3 or L. gasseri BC12 were dose-dependent. At the highest concentration used in our study there were ~1000 EVs per HIV-1 target cell. At this concentration, EVs were not cytotoxic, as evaluated with three different techniques (propidium-iodide-based assay, flow cytometry, and MTT assay). Thus, inhibition of HIV-1 infection was not due to EV-induced cell death.

Not all lactobacilli-released EVs inhibited HIV-1 infection: EVs from L. crispatus BC5 or L. gasseri BC13 did not inhibit HIV-1 infection, although these bacterial strains inhibited HIV-1 replication in human tissues ex vivo[12]. The difference in behavior between EVs released by L. crispatus BC3 and L. gasseri BC12 and between L. crispatus BC5 and L. gasseri BC13 could in principle be due to the different quantity of EVs released by these bacteria or to their qualitative features. We found that there was no difference in either the number or the size distribution between EVs that inhibited and those that did not inhibit HIV-1 infection. Thus, the inhibitory activity of EVs from L. crispatus BC3 and L. gasseri BC12 is not quantitative but rather is related to their composition.

We identified differences in composition between HIV-1-inhibiting and HIV-1-noninhibiting EVs through proteomic and metabolomic analysis. First, the difference in metabolite composition between EVs that inhibit and those that do not inhibit HIV-1 infection was revealed by the principal component analysis of metabolomics: high anti-HIV-1 activity of EVs correlated with their metabolomic profiles. In particular, EVs active toward HIV-1 infection were mainly associated with high amounts of amino acids. Also, our data demonstrated the presence of lactic acid in association with EVs released by lactobacilli. Lactic acid is a metabolite known to exert an anti-HIV-1 effect[4,12,31,32]. The use of lactic-acid-containing diaphragm gel to suppress HIV-1 recently passed Safety Phase I in a pre-clinical study[33].

Proteomic analysis also showed that EVs that inhibited HIV-1 replication differ from those that did not inhibit the virus in terms of several proteins, namely enolase 2, 60 kDa chaperonin, elongation factor Tu, ATP synthase gamma chain, foldase protein PrsA 1, ATP synthase subunit delta, and triosephosphate isomerase. Similar proteins were identified in EVs released by the gastrointestinal bacterium, Lactobacillus casei[22].

Bioactivity of several molecules that we identified in HIV-1-inhibiting EVs was reported earlier: several enolases derived from Lactobacillus were shown to inhibit the adherence of Neisseria gonorrhoeae to epithelial cells[34]. Moreover, bifidobacterial enolase, a cell-surface receptor for human plasminogen, was involved in interaction with human host cells[35]. The elongation factor Tu

was shown to play an important role in the attachment of *Lactobacillus johnsonii* to human intestinal cells and mucins[36]. Whether the inhibition of HIV-1 entry is the result of the action of one or of a combination of several of these bioactive molecules acting synergically remains an open and somewhat difficult question to answer. Also, they may act only when associated with EVs. For these reasons, we tested the effect of EVs as a whole in a cellular model of HIV-1 entry using TZM-bl cells. TZM-bl cells contain integrated reporter genes for firefly luciferase and *E. coli* β-galactosidase under the control of an HIV-1 long terminal repeat[37,38], permitting sensitive and accurate measurements of infection at the entry/attachment level. Here, we showed that viral attachment/entry to TZM-bl cells was inversely proportional to the concentration of bacterial EVs. Another set of experiments performed in the T cell line MT-4 confirmed that EVs directly inhibit HIV-1 attachment to cells. Thus, the anti-HIV-1 effect of *Lactobacillus*-derived EVs is mediated by the decrease of viral entry/attachment to the target cells.

This decrease may be related to the alteration of HIV-1 virions by bacterial EVs. We found that virions pretreated with EVs released by *L. crispatus* BC3 and *L. gasseri* BC12, but not by *L. crispatus* BC5 and *L. gasseri* BC13, were no longer recognized by PG9, an antibody that specifically binds functional trimeric gp120. Bacterial EVs from *L. crispatus* BC3 and *L. gasseri* BC12 interfere with the acessability of viral Env, thus explaining HIV-1 inhibition observed in cell lines and in human tissues ex vivo. In contrast, the pretreatment of cells with bacterial EVs did not affect HIV-1 infection. Bacterial EV-mediated HIV-1 inhibition is therefore the consequence of EVs affecting the infectivity of virions rather than cell functions.

Also, there could be other mechanisms by which EVs inhibit HIV-1 infection. Gram-positive bacteria-derived EVs can change susceptibility of cells to viral infection, e.g., by regulating TLR2 activity[39], by delivering signal molecules into the cells[40–42], or by upregulating host defense genes[24].

Also, bacterial EVs may prevent HIV-1 infection by affecting the vaginal mucosa immune activation status. It has already been demonstrated that the vaginal microbiota, in particular *Lactobacillus* species, affects the immune system in such a way as to efficiently prevent viral infections[13,43–45]. Whether these immunomodulatory effects may be mediated by EVs released from *Lactobacillus* remains to be studied.

In conclusion, in our ex vivo tissue system the protective effect of vaginal *Lactobacillus* against HIV-1 transmission is, in part, mediated by bacterial EVs that inhibit HIV-1 attachment/entry to the target cells due to diminished exposure of viral Env. To extrapolate our findings to the in vivo context, direct experiments in in vivo systems should be performed. These findings may lead to new strategies to prevent male-to-female sexual HIV-1 transmission, for example by use of EVs derived from symbiotic bacteria.

## Methods

**Bacterial cultures and extracellular vesicle isolation**. All *Lactobacillus* strains of vaginal origin, *L. crispatus* BC3, *L. crispatus* BC5, *L. gasseri* BC12, and *L. gasseri* BC13[6], were cultured overnight at 37 °C in tubes containing 50 mL of de Man, Rogosa, and Sharpe (MRS) broth (Difco, Detroit, MI) supplemented with 0.05% L-cysteine. MRS broth was previously autoclaved and filtered with 0.1-μm filters to reduce the amount of particles that might be present in the medium. We achieved anaerobic conditions by cultivating lactobacilli in jars containing Gaspak EZ (BD, Franklin Lakes, NJ). We measured the turbidity of overnight cultures with a spectrophotometer (OD600 nm; Biophotometer, Eppendorf, Germany) and calculated the bacterial concentration using an optical density conversion factor of 0.4, corresponding to a concentration of $10^8$ colony-forming units (CFU) per mL. EVs were isolated by sequential ultracentrifugation[46]. 50 mL of bacterial suspension ($1 \times 10^9$ CFU per mL) was centrifuged at $2500 \times g$ for 10 min at 4 °C. The resulting supernatants were filtered with 0.22-μm filters to eliminate any remaining bacteria. We obtained EVs by ultracentrifuging the samples at $100,000 \times g$ for 70 min at 4 °C

(Ultracentrifuge WX ultra 80, Rotor: AH629 capacity 6 × 36 mL; Thermo Fisher Scientific, Walkersville, MD). The pellet containing bacterial EVs was washed in phosphate-buffered saline (PBS), ultracentrifuged (as specified above), and resuspended in 150 μL of PBS and stored at 4 °C. The same protocol was applied to isolate particles from 50 mL of MRS medium.

**Nanoparticle tracking analysis of bacterial EVs**. Nanoparticle tracking analysis (NTA) utilizes the properties of both light scattering and Brownian motion to obtain the particle size distribution of samples in liquid suspension. Briefly, we diluted EVs 1:100 in PBS and tracked them using the NanoSight NS300 (Malvern instruments Ltd, Malvern, UK) equipped with a 405-nm laser. The samples were flowed by means of a constant pressure syringe pump controller. Videos were recorded for 60 s three times, at camera level 13, and analyzed with NTA software 3.0 (Malvern instruments Ltd, Malvern, UK).

**Cell and human tissue cultures**. Human T-lymphocyte MT-4 (obtained through the NIH AIDS Reagent Program, Germatown, MD, catalog number 120) and Jurkat-tat cell lines (obtained through the NIH AIDS Reagent Program, Germatown, MD, catalog number 1399) were cultured in Roswell Park Memorial Institute (RPMI) 1640 medium (Gibco BRL, Carlsbad, CA) supplemented with 10% heat-inactivated fetal bovine serum (FBS). Jurkat-tat cells were also supplemented with geneticin and hygromycin B, each at 0.5 mg per mL (Thermo Fisher Scientific, Walkersville, MD). TZM-bl cell line (obtained through the NIH AIDS Reagent Program, Germatown, MD, catalog number 8129) were cultured in Dulbecco's modified Eagle's medium (DMEM) supplemented with 10% FBS.

Human tissues were obtained from routine surgery (unrelated to the current study) and according to all relevant ethical regulations for work with human participants including the patient's informed consent. Tonsillectomies were performed in the Children's Hospital (Washington, DC). Tissues were received from the Pathology Department and were considered as "pathological waste". Tissue samples were anonymized and the protocol was approved by the Children's Hospital IRB and by the NIH Office of Human Subject Research. Cervico-vaginal tissues were received as anonymized samples from the National Disease Research Interchange (NDRI). NDRI maintains a Federal Wide Assurance (FWA00006180) agreement with the DHHS, Office for Human Research Protections to comply with federal regulations concerning research involving human subjects. NDRI's human tissue procurement programs and informed consent/authorization documents were approved by the IRB#5 of the University of Pennsylvania.

Tissues were dissected into 2-mm³ blocks, placed onto collagen sponge gel (Gelfoam, Pfizer, New York, NY) at the air–liquid interface, and cultured in RPMI 1640 medium supplemented with FBS at 15%, 1 mM non-essential amino acids, 1 mM sodium pyruvate, amphotericin B at 2.5-μg per mL, and gentamycin sulfate at 50-μg per mL[25].

**Antiviral assays**. Antiviral activities of *Lactobacillus*-derived EVs were investigated in human cell lines and in human tissues ex vivo. A prototypic X4 HIV-1 isolate, LAI.04 (HIV-1$_{LAI.04}$; Rush University Virology Quality Assurance Laboratory, Chicago, IL), and a prototypic R5 HIV-1 isolate, BaL (HIV-1$_{BaL}$; Rush University Virology Quality Assurance Laboratory), were used.

To test the anti-HIV-1 effect in human cell lines, 50 μL of HIV-1$_{LAI.04}$ (stock 350-ng p24$_{gag}$ per mL) or HIV-1$_{BaL}$ (stock 120-ng p24$_{gag}$ per mL) were preincubated with 50 μL of bacterial EVs derived from *L. crispatus* BC3, *L. crispatus* BC5, *L. gasseri* BC12, or *L. gasseri* BC13 (stocks concentrated at $1 \times 10^{10}$ EV per mL), with particles isolated from MRS medium, or with PBS (control), for 1 h, at 37 °C. We then infected $5 \times 10^5$ MT-4 (CXCR4$^+$) or Jurkat-tat (CCR5$^+$) cells with the EV/HIV-1$_{LAI.04}$ or EV/HIV-1$_{BaL}$ mixture, respectively, for 1 h at 37 °C under agitation (400 rpm). After infection, cells were washed with 10 mL of PBS and centrifuged for 5 min at $400 \times g$ to wash off all free virions. The pellet of infected cells was resuspended in 5 mL of RPMI 1640 medium either containing or not containing bacterial EVs ($5 \times 10^8$ particles per mL, corresponding to ~1000 EVs per cell) and transferred to a 24-well plate (1 mL per well; Sigma-Aldrich, St. Louis, MO). MT-4 and Jurkat-tat cells were, respectively, cultured for 3 or 5 days, at 37 °C.

A similar type of experiments as above were conducted to address the concentration-dependent antiviral activity of EVs, choosing EVs released from *L. gasseri* BC12 as a model condition. Cells infected with EV/HIV-1$_{LAI.04}$ or EV/HIV-1$_{BaL}$ were cultured in medium containing or not containing $5 \times 10^4$, $5 \times 10^5$, $5 \times 10^6$, $5 \times 10^7$, or $5 \times 10^8$ bacterial EVs per mL.

For human ex vivo tissues, either HIV-1$_{LAI.04}$ or HIV-1$_{BaL}$ viral stocks were pretreated with *Lactobacillus*-derived EVs, MRS-derived particles, or PBS as mentioned above for 1 h at 37 °C. For HIV-1 infection in human tonsillar tissues ex vivo, nine blocks per well were placed on collagen sponge gel and infected with 15 μL of EV/ HIV-1 or EV-free/HIV-1 mixture on the top of each block. HIV-1$_{BaL}$-infected tonsillar tissues were cultured in tissue culture medium containing or not containing $5 \times 10^8$ EVs per mL derived from *L. crispatus* (BC3 or BC5) and *L. gasseri* (BC12 or BC13). In contrast, cervico-vaginal ex vivo tissues were infected with 500 μL of EV/HIV-1$_{BaL}$ or EV-free/HIV-1 mixture for 2.5 h at 37 °C in agitation. Afterward, the infected cervico-vaginal tissue blocks were washed three times with PBS and transferred at the liquid–air interface onto Gelfoam (nine

blocks per well). All tissue cultures ex vivo were kept for 12 days at 37 °C, replacing with the medium containing or not containing bacterial EVs ($5 \times 10^8$ particles per mL) every 3 days.

The concentration-dependent antiviral activity of EVs derived from *L. gasseri* BC12 was tested in tonsillar and cervico-vaginal ex vivo tissues. Tissues infected with HIV-1$_{LAI.04}$ or HIV-1$_{BaL}$ were cultured in tissue culture medium containing tenfold serial dilutions of bacterial EVs ($5 \times 10^8$, $5 \times 10^7$, $5 \times 10^6$, and $5 \times 10^5$ particles per mL). We used eighteen tissue blocks per each condition (nine tissue blocks per well).

In all the experiments, we evaluated HIV-1 replication by measuring the p24$_{gag}$ antigen released in tissue culture medium, using an immunofluorescent cytometric bead-based assay by Luminex[47].

**Viral entry/cell attachment assays**. HIV-1 viral entry was evaluated in TZM-bl cells (obtained through the NIH AIDS Reagent Program, MD). We cultured $2 \times 10^4$ TZM-bl cells per well in 100 μL of complete DMEM medium in a 96-well black plate (Sigma-Aldrich, St. Louis, MO) for 6 h to ensure complete cell adhesion. Afterward, the medium was replaced with 90 μL of fresh medium containing diverse concentrations of EVs ($5 \times 10^5$, $5 \times 10^6$, $5 \times 10^7$, or $5 \times 10^8$ particles per mL) derived from *L. gasseri* BC12, or $5 \times 10^8$ particles per mL of particles derived from MRS medium, and the cells were infected with 10 μL of HIV-1$_{LAI.04}$ stock, overnight. After infection, the medium was replaced with medium containing different amounts of bacterial EVs and cultured at 37 °C for 3 days. To measure HIV-1$_{LAI.04}$ entry/integration to the cell, we removed the cell culture medium and lysed the cells with 100 μL of 1X of Glo lysis buffer (Promega, Madison, WI). We added 100 μL of 2X substrate Bright Glo luciferase buffer (Promega, Madison, WI). After a 3-min incubation, we measured the luminescence with a Saphire 2 luminometer (Tecan, Switzerland).

To study viral attachment/entry to cells, we incubated 150 μL of MT-4 cell suspension ($5 \times 10^6$ cells) with 150 μL of HIV-1$_{LAI.04}$ in medium containing or not containing $5 \times 10^8$ EVs released by *L. crispatus* BC3, *L. crispatus* BC5, *L. gasseri* BC12, *L. gasseri* BC13, or particles derived from MRS medium for 2 h at 4 °C. Then, the infected cells were washed extensively five times with 1 mL of RPMI medium to eliminate all free viral particles by centrifugation at $400 \times g$ for 5 min. Afterward, cells were lysed with lysis buffer (PBS containing 1% Triton X-100, 20 mM tris HCl, 0.05% tween-20, and 30% bovine serum albumin), and viral attachment/entry into the cell was evaluated in the cell lysates from measurements of the viral p24$_{gag}$ protein with Luminex.

To study whether bacterial EVs enter into the host cells and alter host cell function to induce protection against HIV-1, either TZM-bl or MT-4 cells were treated with bacterial EVs derived from *L. crispatus* BC3, *L. crispatus* BC5, *L. gasseri* BC12, *L. gasseri* BC13, and MRS-derived particles for 24 h, prior to HIV-1 infection, and then HIV-1 infectivity was evaluated. For TZM-bl cells, $2 \times 10^4$ cells per well were cultured in 100 μL of complete DMEM medium in a 96-well black plate (Sigma-Aldrich, St. Louis, MO) containing or not containing bacterial EVs at a concentration of $5 \times 10^8$ EV per mL, at 37 °C for 24 h. Afterward, the cells were infected with 10 μL of HIV-1$_{LAI.04}$ stock and cultured at 37 °C for 3 days. To measure HIV-1$_{LAI.04}$ replication, we removed the cell culture medium and lysed the cells with 100 μL of 1X of Glo lysis buffer (Promega, Madison, WI). We added 100 μL of 2X substrate Bright Glo luciferase buffer (Promega, Madison, WI). After a 3-min incubation, the luminescence was measured with a Saphire 2 luminometer (Tecan, Switzerland). For MT-4 cells, $1 \times 10^5$ per well were cultured in 1 mL of complete RPMI medium containing or not containing EVs in a 24-well plate (Sigma-Aldrich, St. Louis, MO), at 37 °C for 24 h. Then, MT-4 cells were infected with 10 μL of HIV-1$_{LAI.04}$ stock and cultured at 37 °C for 3 days. We evaluated HIV-1$_{LAI.04}$ replication by measuring the p24$_{gag}$ antigen released in the cell culture medium (Luminex).

**HIV-1 capture using magnetic nanoparticles**. Human monoclonal PG9 antibody (1 mg per mL; Polymun Scientific, Austria, AB015), which preferentially recognizes trimeric HIV-1 envelope protein (gp120), was coupled to 15-nm carboxyl-terminated magnetic iron oxide nanoparticles (MNPs) according to the manufacturer's protocol (Ocean NanoTech, San Diego, CA).

To study the effect of *Lactobacillus*-derived EVs on HIV-1 itself, we pretreated 50 μL of HIV-1$_{LAI.04}$ viral particles (stock 350-ng per mL p24$_{gag}$) with 50 μL of bacterial EVs derived from *L. crispatus* BC3, *L. crispatus* BC5, *L. gasseri* BC12, and *L. gasseri* BC13 (stocks concentrated at $1 \times 10^{10}$ EV per mL), with particles isolated from MRS medium or with PBS (control), for 1 h at 37 °C.

Afterwards, HIV-1 viral particles were captured with 50 μL of PG9-MNP-anti-gp120 antibody (stock 1 mg per mL) for 1 h, at 37 °C. We then separated HIV-1 virions captured by MNPs from the non-captured ones using magnetic columns attached to a high field MACS magnet (Miltenyi Biotech, Auburn, CA). After washing columns three times with 600 μL of washing buffer (0.5% bovine serum albumin, 1 mM ethylenediaminetetraacetic acid), we removed the MNP–HIV-1 complexes from the magnet, demagnetized them for 5 min, and eluted them with 600 μL of lysis buffer (PBS containing 1% Triton X-100), and measured the concentrations of HIV-1 p24$_{gag}$ antigen using an immunofluorescent cytometric bead-based assay by Luminex.

**Cell viability and cell depletion assays**. We performed cell viability assays in MT-4 and Jurkat-tat cells using the NucleoCounter® NC-100™ automated cell counting system (ChemoMetec, Denmark). We determined the numbers of total and dead cells in control cultures and in each *Lactobacillus*-derived EV-treated cultures (*L. crispatus* BC3, *L. crispatus* BC5, *L. gasseri* BC12, *L. gasseri* BC13, $5 \times 10^8$ particles per mL) after 3 days of culture, using a propidium-iodide-based assay according to the manufacturer's protocol (ChemoMetec, Denmark). We also tested the effect of particles derived from MRS bacterial medium ($5 \times 10^8$ particles per mL) as described above. Triton X-100 at 0.2% was used in our study as a potent cytotoxic agent.

We also measured cell viability using the MTT (3-(4,5-dimethylthiazol-2-yl)-2,5-diphenyltetrazolium bromide) tetrazolium reduction kit according to the manufacture's protocol (Sigma-Aldrich, St. Louis, MO). The MTT assay measures the mitochondrial dehydrogenase activity of living cells. Briefly, after day 3 of cell cultures (treated or not treated with EVs), 10 μL of MTT solution was added to each well (10% of culture volume) and plates were incubated for 4 h at 37 °C. After the incubation period, to dissolve the formazan crystals we added MTT solvent at 100 μL per well and incubated the plates for 1 h. After ensuring that the formazan crystals were dissolved, we measured the absorbance at 570 nm on Tecan Sapphire 2 using Magellan 5.0 software (Tecan Group Ltd., Switzerland).

We evaluated cell depletion on MT-4 cells treated or not treated with bacterial EVs (derived from *L. gasseri* BC12, $5 \times 10^8$ particles per mL) using flow cytometry. After 3 days of culture, the cells were centrifuged at $400 \times g$, resuspended in 1 mL of staining buffer (PBS containing 2% mouse serum, 2% goat serum, and 2% FBS), and stained with 1 μL of live/dead Fixable Viability Dye eFluor 450 (ef 450, Invitrogen, Carlsbad, CA) for 20 min. After incubation, cells were washed and further diluted with staining buffer and stained with anti-CD3-AF488 (Thermo Fisher Scientific, Waltham, MA, MHCD0320) and anti-CD4-BV605 (BD Biosciences, San Jose, CA, 562658) fluorescence-labeled monoclonal antibodies for 20 min. Data were acquired with a Novocyte flow cytometer (ACEA Biosciences, CA) equipped with 405, 488, and 650 nm laser lines. We analyzed the data using NovoExpress version 1.2.4 software (ACEA Biosciences, CA).

**Protein extraction, SDS-PAGE, and western blotting**. Total proteins were extracted either from bacterial EVs or from HIV-1-infected/non infected MT-4 cells treated or not treated with bacterial EVs, using RIPA Lysis and Extraction Buffer (Thermo Fisher Scientific, Waltham, MA). We loaded 10 μg of proteins on a 4–20% precast polyacrylamide gel (Bio-Rad Laboratories, Hercules, CA), separated them using SDS-PAGE, and then transferred them to low-fluorescence PVDF membranes and probed them with anti-HIV-1 p24 (0.5 μg per mL; Abcam, Cambridge, MA, ab9071), anti-p53 (1 μg per mL; Thermo Fisher Scientific, Waltham, MA, MA5-12557), anti-CD63 (1 μg per mL; Thermo Fisher Scientific, Waltham, MA, 10628D), anti-TSG101 (1 μg per mL; Thermo Fisher Scientific, Waltham, MA, MA1-23296), primary anti-mouse monoclonal antibodies, and then goat peroxidase-conjugated anti-mouse IgG secondary antibody (Bio-Rad Laboratories, Hercules, CA). We detected peroxidase activity and digital images using the ChemiDoc™ MP Imaging System (Bio-Rad Laboratories, Hercules, CA).

**¹H-NMR analysis**. We subjected 500 μL of EVs (stocks concentrated at $5 \times 10^{10}$ particles per mL) released by *L. crispatus* BC3, *L. crispatus* BC5, *L. gasseri* BC12, or *L. gasseri* BC13, or particles isolated from MRS medium, to a water bath sonication for 15 min to ensure complete EV lysis preceding ¹H-NMR spectroscopy analysis. ¹H-NMR spectra were recorded at 298 K with an AVANCE spectrometer (Bruker, Milan, Italy) operating at a frequency of 600.13 MHz, equipped with an autosampler with 60 holders[48]. Each spectrum was acquired using 32 K data points over a 7211.54-Hz spectral width and adding 256 transients A recycle delay of 5 s and a 90° pulse of 11.4 s were set up. The signals originating from large molecules were suppressed by a CPMG filter of 400 echoes, generated by 180° pulses of 24 μs separated by 400 μs. The signals were assigned by comparing their multiplicity and chemical shift with Chenomx software data bank (ver 8.1 Chenomx, Inc., Edmonton, Canada). We looked for features of the metabolome related to antiviral activity by evaluating the Pearson correlation between such activity and the concentration of each molecule identified. For this purpose, the function corr.test from the R (www.r-project.org) was employed. To highlight any overall trend characterizing the samples, we used the molecules whose concentration was significantly correlated to antiviral activity ($p < 0.05$) as a basis for a robust principal component analysis (rPCA) model[49]. For each model, we calculated the scoreplot, i.e., the projection of the samples in the principal components (PC) space, tailored to highlight the underlying structure of the data. Besides, we ranked the importance of each molecule in determining each PC. We did this by calculating how the coefficients attributed to each of them to determine each PC correlated, according to Pearson, with its concentrations.

**Mass spectrometry of bacterial EVs**. EVs (50 μL, from stocks at $5 \times 10^{10}$ particles per mL) isolated either from *L. gasseri* BC12 or *L. crispatus* BC5 were analyzed for proteomics with mass spectroscopy (NICHD Biomedical Mass Spectrometry Facility, Bethesda, MD). Frozen EV samples were suspended in 0.1 M ammonium bicarbonate and solubilized with 0.2% ProteaseMAX (Promega Corp, Madison, WI) surfactant. Proteins were reduced with dithiothreitol and alkylated with iodoacetamide, before digestion overnight with sequencing grade trypsin (Promega Corp., Madison, WI). The digestion was quenched with 1 volume of 5% formic

acid:acetonitrile (1:1), dried under vacuum, and reconstituted in 0.1% formic acid: acetonitrile (1:1). We analyzed the resulting peptides with LC-ESI-MS/MS using a 6560 IM Q-TOF mass spectrometer (Agilent Technologies, Santa Clara, CA) equipped with a Jet Stream Electrospray ionization source. We injected and separated aliquots of digested samples using reversed phase C18 gradient chromatography and collected MS/MS fragment spectra in a data dependent mode. We extracted tandem mass spectra from each run in Mascot Generic Format (MGF) and analyzed them using Mascot (Matrix Science, London, UK; version 2.6.2) searching the Eubacteria subset of the Sprot 2018_11 protein database (333,732 sequences) and a reverse sequence decoy database to determine the false discovery rate (FDR). Digestion enzyme specificity was set for trypsin with a maximum of 2 missed cleavages. The parent and fragment mass tolerances were set at 5 ppm and 0.05 Da respectively; carbamidomethyl (C) was set as a fixed and oxidation (M) as a variable modification. Scaffold (ver 4.8.7; Proteome Software Inc., Portland, OR) was used to validate MS/MS based peptide and protein identifications[50]. Peptide identifications were accepted if they could be established at >95.0% probability as specified by Peptide Prophet[51]. We assigned protein probabilities using the Protein Prophet algorithm (Nezvizhskii, 2003) and accepted protein identifications if they could be established at a 5% FDR and inclusive of at least two identified peptides. We obtained GO annotations for identified proteins using Scaffold with data imported from the NCBI website.

**Statistical analysis**. We performed all statistical analyses, unless otherwise specified, using the ordinary one-way ANOVA test (multiple comparison with Dunnett's correction; GraphPad Prism version7; GraphPad Prism Software Inc., San Diego, CA). Results were expressed as means ± standard error of mean, and differences were deemed significant for $p$-values < 0.05.

**Reporting summary**. Further information on research design is available in the Nature Research Reporting Summary linked to this article.

## Data availability

The data that support the findings of this study are available from the corresponding author L.M. upon reasonable request. The source data underlying Figs. 1b–d, 2a–c, 3a–b, 4a–c, 5a–b, 6, 7a–b, 9a–b, and Supplementary Fig. 1, are provided as a Source Data file. The mass spectrometry data have been deposited to the Mass spectrometry Interactive Virtual Environment (MassIVE) database (dataset identifier MSV000084514).

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

## Acknowledgements

This work was supported by the Intramural Research Program of the Eunice Kennedy Shriver National Institute of Child Health and Human Development, National Institutes of Health. We thank the Department of Pathology of Children's National Medical Center for providing us with human tonsillar tissues.

## Author contributions

L.M. and C.V. designed experiments, analyzed the data, and wrote the paper. R.A.Ñ.P. designed and performed the experiments, analyzed the data, and wrote the paper. L.L., C.P., and B.V. led the metabolome data generation. P.B. generated the proteomic data. K.M. performed experiments. All the authors contributed to data interpretation. All the authors read, reviewed, and approved the final paper.

## Competing interests

The authors declare no competing interests.
