## [Peer Review File · Nature Communications]

Reviewers' comments:

Reviewer #1 (Remarks to the Author):

The current manuscript by Palomino and Margolis lab describes the effect of EVs from vaginal lactobacilli inhibiting HIV-1 in tissues. The authors focus on the vaginal microbiota, dominated mostly by *Lactobacillus* spp, which plays a key role in defending the female genital tract against numerous urogenital pathogens, including HIV-1. Data shows that they were able to isolate EVs derived from four different strains of *Lactobacillus* all of which were capable of inhibiting HIV-1 infection. Inhibition of HIV-1LAI infection of MT-4 cells by EVs was concentration-dependent and the effect progressively augmented with increase in the concentration of bacterial EVs. Finally, some of the regulation may be TLR2 dependent. Overall, this is an interesting and exciting manuscript and the fact that they used tissue explants (data in Figure 4) adds a great deal of relevance, since they are primary cells. There are some suggestions that may improve the manuscript including:

1. Do the authors have a sense of the ratio of EV to cells that were used in some of their assays?
2. Are the bacterial EV uptake any different from human cell EVs? Can one stain the EVs and look for uptake kinetics in infected vs. uninfected cells.
3. PI staining in page 7 for the bacterial EV not being toxic may not be sufficient and requires 1-2 more assays?
4. Are p53 or p21/waf1 altered following bacterial EV treatment of infected or uninfected cells.
5. Data in figure 6 is exciting but needs better explanation of how it can contribute to HIV regulation in infected cells (presumably T-cells and/or myeloids in tissues)?

Reviewer #2 (Remarks to the Author):

The microbiome is strongly linked to increased HIV susceptibility in both women and men, and this paper by Palomino et al studies a very interesting aspect of the microbiome in HIV infection - the role of extracellular vesicles. To date much of the studies of the vaginal microbiome in HIV infection in women has focused on compositional characterization of bacteria by molecular or microscopy techniques, linking vaginal Lactobacillus species with lower risk of HIV acquisition. However, there has been a significant gap in knowledge in how these bacteria may contribute to this observation from multiple clinical cohort studies. Much of prior studies have looked at anti-inflammatory effects of Lactobacillus species on vaginal epithelial cells, how metabolites affect inflammation responses, antimicrobial factors such as H2O2 from bacteria, etc. This study examines the role of extracellular vesicles (EV's) produced by clinically isolates of *L. crispatus* and *L. gasseri* made in bacterial culture. By using standard in vitro HIV inhibition assays of X and R-tropic strains in reporter cell lines they show that EV's isolated from these cultures can inhibit virus production, which they validated in a more physiologically relevant explant model of cervical tissue. There is some evidence to suggest this mechanism of inhibition may be mediated at the viral entry/attachment step. The paper provides data on the compositional nature of these EV's, which contain many bacterial proteins as well as metabolites, although it is not explored which of these may be mediating activity (alone or in combination). Therefore, this paper provides novel information on the functionality of the vaginal microbiome that may be important for host immunity against viruses.

The data, however, does not support the conclusion in the abstract, or the final paragraph of the discussion, that EV's provide protective capacity against HIV in vivo, as none of these experiments showed any data to support relevance for in vivo activity. While the in vitro data is compelling and hypothesis-generating, it is a large leap to come to this conclusion. For example, what is the physiologically concentration of Lactobacillus related EV's in cervicovaginal mucus of women? Does it even reach levels tested in this paper? While many of these proteins and/or metabolites may have been characterized in cervicovaginal mucus from other studies, mucosal samples from women were not presented in this paper. This might be a challenging thing to tease out relative contributions of various bacterial species to EV's that would presumably be present in the female genital tract and/or host cells. In any case, would they be at a protective level that would be helpful against HIV exposure, in line with the 10^8 bacterial EV/ml used in these experiments?

Also, in the discussion, is there anything known about how EV's might inflammation responses in the vagina or other mucosal surfaces? Elevated levels of pro-inflammatory cytokines are also a major risk factor for HIV, and this would be important to present a discussion in the context in relation to EV's from bacteria in general.

Figure 5A: How reproducible is this data? There are 3 technical replicates mentioned, one of which shows 2 data points (5×10^7 column). It is unclear whether this was simply one experiment done in triplicate, or this was 3 x 3 triplicate experiments showing the means of each, in line with the other experiments presented in the manuscript.

Reviewer #3 (Remarks to the Author):

Nahui Palomino and colleagues hypothesized that extracellular vesicles (EVs) from vaginal *Lactobacillus* spp. may contribute to the reported protective effects of these bacteria against HIV-1 transmission in the female genital tract. The authors found that the EVs from two vaginal spp., *L. crispatus* BC3 and *L. gasseri* BC12, were particularly effective at protecting cell lines and human tissue models against HIV-1 infection. This protection was reported to be associated with the ability of the EVs to affect viral attachment to cells and to correlate with several EV metabolic and protein components.

This is an interesting piece of work with potentially important ramifications for the field of HIV-1 infection. The work also reveals a new aspect of bacterial EV biology by showing how bacterial vesicles may interfere with the infectious process of viruses and, possibly, other pathogens. Nevertheless, the work is incomplete and also inconclusive with respect to the mode of action of the *Lactobacillus* EVs on viral infection, leaving two key questions left unanswered. Firstly, do the EVs affect viral infectivity? i. e. do the EVs have any effects on the surface or other properties of the virions that might alter their ability to adhere to and enter host cells? If the HIV-1 particles are pre-incubated with the EVs, then added to cells, do the authors also observe decreased viral replication? A second question is whether the EVs enter host cells and, if so, may affect host cell functions to prevent HIV adherence and/or entry. Although it is unlikely that many EVs would be able to enter in one hour, this question needs to be addressed experimentally.

The statistical analyses appear appropriate. Other specific points for the authors' consideration are listed below:

Major points:

- 1) Line 83: The authors describe the presence of EVs in the MRS medium. Other than the average size of these EVs, no further information is provided. It is likely that the EVs present in the medium are of eukaryotic origin, however, as these have been used as controls in some experiments, the authors need to characterize them further.
- 2) Line 124 Fig. 3): A positive control is required to show that cell death can be detected in this model.
- 3) Lines 139-143 (Fig. 4a): MRS EVs also need to be tested in this cell line, as in Fig. 2a.
- 4) Lines 163-166 (Fig. 5a): MRS EVs need to be tested in this cell line.

5) Lines 167-172 (Fig. 5b): Do the BC3 EVs also block HIV-1 cell attachment, whereas those of BC5, BC13 and MRS do not?

6) Lines 185-214: The data from the metabolic and proteomic studies are not convincing. Although a PCA of the metabolite composition of EVs identified some trends between the EVs that blocked HIV-1 entry and those that didn't, the data are based on only 5 different samples, therefore making it impossible to reach any firm conclusions. Similarly, the proteomic analyses have really only identified proteins that are commonly found in bacterial EV preparations. The authors would need to show that the inactivation of one or more of the corresponding genes had an effect on the protective activity of the EVs. Alternatively, they would need to perform proteomic analyses on multiple samples (biological replicates) for each type of EV to show that the trends are reproducible and/or analyze more types of EVs. Related to this point, were multiple batches of EVs for each *Lactobacillus* spp. prepared and tested or were biological replicates of cell culture studies performed with just one batch of each type of EV?

Minor points:

1) Line 69: Reiterate here the reason here for the focus on vaginal lactobacilli and briefly mention the reason(s) for choosing these particular strains.

2) Lines 90-91: More information is required here for the non-specialist reader.

3) Line 113: Why were tonsillar tissues used?

POINT-BY-POINT RESPONSE TO THE REFEREES' COMMENTS

In the point-by-point response, line numbers are referred to the clean revised version.

Response to Reviewer 1:

Reviewer #1 (Remarks to the Author):

The current manuscript by Palomino and Margolis lab describes the effect of EVs from vaginal lactobacilli inhibiting HIV-1 in tissues. The authors focus on the vaginal microbiota, dominated mostly by Lactobacillus spp, which plays a key role in defending the female genital tract against numerous urogenital pathogens, including HIV-1. Data shows that they were able to isolate EVs derived from four different strains of Lactobacillus all of which were capable of inhibiting HIV-1 infection. Inhibition of HIV-1LAI infection of MT-4 cells by EVs was concentration-dependent and the effect progressively augmented with increase in the concentration of bacterial EVs. Finally, some of the regulation may be TLR2 dependent. Overall, this is an interesting and exciting manuscript and the fact that they used tissue explants (data in Figure 4) adds a great deal of relevance, since they are primary cells. There are some suggestions that may improve the manuscript including:

We thank the reviewer for the overall positive evaluation of our work and for emphasizing the importance of human tissues *ex vivo* to study HIV-1 pathogenesis.

1. Do the authors have a sense of the ratio of EV to cells that were used in some of their assays?

Yes, we do. In anti-HIV-1 experiments performed in cell lines, at the highest concentration of EVs, 5×10^8 EVs were added to 5×10^5 cells and therefore the ratio was approximately 1000 EVs per cell. In experiments with tissue explants, 9 tissue blocks are typically cultured in 3 mL of medium. Therefore, 1.5×10^9 EVs were used (5×10^8 EVs/mL \times 3) for a total of approximately 1.08×10^6 CD4+ T cells (1.2×10^5 CD4+ T cells per block \times 9 blocks) (Grivel, 2009), thus giving a ratio of 1/1000 as well. Although for tissue blocks it is difficult to calculate ratios precisely, the numbers of EVs per target cell were approximately the same in both experimental systems used in our study. We added this information in the manuscript (see lines 124-126 and lines 191-194).

2. Are the bacterial EV uptake any different from human cell EVs? Can one stain the EVs and look for uptake kinetics in infected vs. uninfected cells.

This is an interesting question. We performed the experiments suggested by the reviewer using confocal microscopy. We incubated TZM-bl cells with bacterial EVs stained with Bodipy FL for 2 h, and then we stained cells with the fluorescent membrane dye FM4-64. We found that some EVs were inside the cells and some on the cell surfaces. We present several confocal images below for illustration. To test whether there are some subtle differences in location of the EVs inside the cells, more systemic studies are needed. Unless the reviewer insists, we decided not to

include these data in our manuscript, especially since we have now generated additional proofs that EVs act at the “virus level” by preventing interaction between gp120 and cells.

Internalization of BODIPY labelled EVs (shown in green) into cell after 2 h incubation. Plasma membrane is stained with FM4-64 (shown in red) to delineate cell boundaries. Images show untreated control, BC03, BC05 (top panels from left to right), BC12, BC13 and MRS (bottom panels from left to right). Green Arrow shows an example of EV internalization. Yellow arrow shows an example of cell surface bound EV. Scale bar is 10 μ M.

3. PI staining in page 7 for the bacterial EV not being toxic may not be sufficient and requires 1-2 more assays?

We followed the reviewer’s request and performed new experiments. In the original manuscript, we reported on the lack of toxicity of bacterial EVs in two cell lines (MT-4 and Jurkat-tat) using two techniques (propidium-iodide based assay and flow cytometry). However, to answer the reviewer’s request, we used an MTT assay to evaluate cell metabolic activity (see Fig. 3b and Methods section, lines 564-572). No toxicity was observed in any of the three different assays used to evaluate cell viability in two different cell lines; thus toxicity did not account for bacterial EV-mediated HIV-1 inhibition. Triton at 0.2% was used as control for toxicity measurement. We present these results on lines 139-158 and discuss them on lines 341-342.

4. Are p53 or p21/waf1 altered following bacterial EV treatment of infected or uninfected cells.

To address the reviewer’s question, we performed new experiments. We infected MT-4 cell cultures with X4_{LAI.04}, treated them or not with bacterial EVs, and evaluated p53 expression by

Western blot. In agreement with several studies showing that HIV-1 infection induces expression of p53 in primary CD4+ T cells (Imbeault, 2009; Genini, 200; Imbeault, 2009), we observed here the expression of p53 in all cultures infected with HIV-1. Nevertheless, we did not observe visible increase/decrease of p53 protein levels upon cell treatment with bacterial EVs (see Supplementary Figure 1). These results are reported in of the amended version of our manuscript on lines 127-128.

5. Data in figure 6 is exciting but needs better explanation of how it can contribute to HIV regulation in infected cells (presumably T-cells and/or myeloids in tissues)?

We thank the reviewer for this remark. In the amended version of our manuscript we explained how the metabolites associated to *Lactobacillus*-derived EVs may contribute to HIV-1 regulation in infected cells. We found significantly higher amounts of glutamate, hypoxanthine, glycine, and methionine associated to bacterial EVs that decreased HIV-1 replication (previously Figure 6; now Figure 8) These data are the basis for evaluating the potential role of these metabolites or their combinations in regulating HIV-1 infection. Unless all reviewers think that these data are irrelevant to this current study, we would like to keep them in the paper.

Also, our data (Supplementary Table 1) demonstrated the presence of lactic acid in association with EVs released by lactobacilli. Lactic acid is a metabolite known to exert an anti-HIV-1 effect (O'Hanlon, 2011, 2013; Nahui, 2017; Tyssen, 2018). The use of lactic-acid-containing diaphragm gel to suppress HIV-1 recently passed Safety Phase I in a pre-clinical study (Thurman, 2019). We now discuss this on lines 358-361.

Reviewer #2 (Remarks to the Author):

The microbiome is strongly linked to increased HIV susceptibility in both women and men, and this paper by Palomino et al studies a very interesting aspect of the microbiome in HIV infection - the role of extracellular vesicles. To date much of the studies of the vaginal microbiome in HIV infection in women has focused on compositional characterization of bacteria by molecular or microscopy techniques, linking vaginal Lactobacillus species with lower risk of HIV acquisition. However, there has been a significant gap in knowledge in how these bacteria may contribute to this observation from multiple clinical cohort studies. Much of prior studies have looked at anti-inflammatory effects of Lactobacillus species on vaginal epithelial cells, how metabolites affect inflammation responses, antimicrobial factors such as H2O2 from bacteria, etc. This study examines the role of extracellular vesicles (EV's) produced by clinically isolates of L. crispatus and L. gasseri made in bacterial culture. By using standard in vitro HIV inhibition assays of X and R-tropic strains in reporter cell lines they show that EV's isolated from these cultures can inhibit virus production, which they validated in a more physiologically relevant explant model of cervical tissue. There is some evidence to suggest this mechanism of inhibition may be mediated at the viral entry/attachment step. The paper provides data on the compositional nature of these EV's, which contain many bacterial proteins as well as metabolites, although it is not explored which of these may be mediating activity (alone or in combination). Therefore, this paper provides novel information on the functionality of the vaginal microbiome that may be important for host immunity against viruses.

We thank the reviewer for the generally positive evaluation of our work and for emphasizing that our paper provides new information on the functionality of the vaginal microbiome in fighting pathogens.

The data, however, does not support the conclusion in the abstract, or the final paragraph of the discussion, that EV's provide protective capacity against HIV in vivo, as none of these experiments showed any data to support relevance for in vivo activity. While the in vitro data is compelling and hypothesis-generating, it is a large leap to come to this conclusion. For example, what is the physiologically concentration of Lactobacillus related EV's in cervicovaginal mucus of women? Does it even reach levels tested in this paper? While many of these proteins and/or metabolites may have been characterized in cervicovaginal mucus from other studies, mucosal samples from women were not presented in this paper. This might be a challenging thing to tease out relative contributions of various bacterial species to EV's that would presumably be present in the female genital tract and/or host cells. In any case, would they be at a protective level that would be helpful against HIV exposure, in line with the 10^8 bacterial EV/ml used in these experiments?

We thank the reviewer for this constructive criticism. Although in our *in vitro* and *ex vivo* experiments we used EVs released by lactobacilli isolated from healthy women (Parolin, 2015), and these EVs exhibit anti-HIV-1 activity at a wide range of concentrations, we do not know, as the reviewer correctly pointed out, what the concentration of EVs in the vagina is and what the role of cervico-vaginal mucus with regard to these EVs is. In the amended version of our paper we corrected both the abstract and the final paragraph, emphasizing that our data with *ex vivo* tissues cannot be directly extrapolated to the situation *in vivo* (see lines 27-30 and lines 401-407).

Also, in the discussion, is there anything known about how EV's might inflammation responses in the vagina or other mucosal surfaces? Elevated levels of pro-inflammatory cytokines are also a major risk factor for HIV, and this would be important to present a discussion in the context in relation to EV's from bacteria in general.

We thank the reviewer for bringing up an interesting hypothesis regarding the relation of EVs to inflammation and in particular to pro-inflammatory cytokines. As the reviewer suggested we now present a discussion on the relation between bacterial EVs and inflammation. In general, little is known about the relationship between the vaginal microbiota and the immune system and hence even less about the role of EVs in these processes. Nevertheless, it has already been demonstrated that the vaginal microbiota, in particular *Lactobacillus* species, could regulate and stimulate the immune system to efficiently prevent viral infections (Petrova, 2013; Rose, 2012; Wagner, 2012; Karlsson, 2012). Whether these immunomodulatory effects may be mediated by EVs released from *Lactobacillus* has to be studied. These points are now discussed on lines 396-400 of our manuscript.

Figure 5A: How reproducible is this data? There are 3 technical replicates mentioned, one of which shows 2 data points (5×10^7 column). It is unclear whether this was simply one experiment done in triplicate, or this was 3×3 triplicate experiments showing the means of each,

in line with the other experiments presented in the manuscript.

We apologize for the lack of clarity in Figure 5a. Our results are presented as means of independent experiments. We decided to perform more experiments to increase the reproducibility of our data. We have now added this information (see Fig.5a, lines 197-204).

Reviewer #3 (Remarks to the Author):

*Nahui Palomino and colleagues hypothesized that extracellular vesicles (EVs) from vaginal *Lactobacillus* spp. may contribute to the reported protective effects of these bacteria against HIV-1 transmission in the female genital tract. The authors found that the EVs from two vaginal spp., *L. crispatus* BC3 and *L. gasseri* BC12, were particularly effective at protecting cell lines and human tissue models against HIV-1 infection. This protection was reported to be associated with the ability of the EVs to affect viral attachment to cells and to correlate with several EV metabolic and protein components.*

*This is an interesting piece of work with potentially important ramifications for the field of HIV-1 infection. The work also reveals a new aspect of bacterial EV biology by showing how bacterial vesicles may interfere with the infectious process of viruses and, possibly, other pathogens. Nevertheless, the work is incomplete and also inconclusive with respect to the mode of action of the *Lactobacillus* EVs on viral infection, leaving two key questions left unanswered. Firstly, do the EVs affect viral infectivity? i. e. do the EVs have any effects on the surface or other properties of the virions that might alter their ability to adhere to and enter host cells? If the HIV-1 particles are pre-incubated with the EVs, then added to cells, do the authors also observe decreased viral replication? A second question is whether the EVs enter host cells and, if so, may affect host cell functions to prevent HIV adherence and/or entry. Although it is unlikely that many EVs would be able to enter in one hour, this question needs to be addressed experimentally.*

We thank the reviewer for the positive evaluation of our work on the role of vaginal microbiota-derived EVs in HIV-1 infection. Also, we thank the reviewer for formulating critical questions regarding the mode of action of *Lactobacillus*-derived EVs. To answer these questions we performed new experiments and amended the text accordingly. We think that we have answered the reviewer's questions.

To answer the first question, we pre-treated HIV-1 with *Lactobacillus*-derived EVs for 1 h. Then we probed HIV-1 envelope proteins with magnetic nanoparticles (MNPs) coupled to PG9 antibodies that preferentially recognize the HIV-1 trimeric envelope proteins, gp120. We isolated virions captured by PG9-MNPs using magnetic columns and quantified them by measuring p24 (see Methods section, lines 536-553). We found that the pretreatment of HIV-1 with *Lactobacillus*-derived EVs, in particular *L. crispatus* BC3- and *L. gasseri* BC12-derived EVs, significantly decreased the number of viral particles captured by PG9-MNPs, thus demonstrating that they prevent the binding of gp120 to PG9. In contrast, neither the pretreatment with control medium-derived EVs nor the pre-treatment with BC5- and BC13-derived EVs prevented the binding of gp120 to PG9 (the numbers of viral particles captured by PG9-MNPs with or without pre-treatment with BC5- and BC13-derived EVs were the same). These results show that that the

numbers of accessible functional Env (trimeric gp120) on virions treated specifically with BC3- and BC12-derived EVs (the ones inhibiting HIV) were significantly decreased. These results are in agreement with the inhibition of HIV-1 replication by BC3- and BC12-derived EVs but not by BC5- or BC13-derived EVs. More specifically, these results show that decreased viral attachment/entry into host cells accounts for the general reduction of HIV-1 replication after pretreatment of HIV-1 with bacterial EVs cells reported in our original paper. We present these results on Fig.6, lines 217-231.

To answer the second critical point raised by the reviewer, namely the recommendation to study whether bacterial EVs enter into host cells and alter host cell functions to induce protection against HIV-1, we treated either TZM-bl or MT-4 cells with bacterial EVs for 24 h prior to HIV-1 inoculation and then evaluated HIV-1 infection (see Methods section, lines 519-534). Our results showed that cell pretreatment with bacterial EVs for 24 h did not significantly suppress HIV-1 replication in either TZM-bl or MT-4 cells. Therefore, of the two potential modes of actions suggested by the reviewer, we rejected the possibility that EVs affect HIV-1 target cells and proved instead that bacterial EVs affect virions' infectivity, thus resulting in HIV-1 inhibition. We present these results in Fig. 7, lines 233-244 and discuss them on lines 384-391. We thank the reviewer for the clear formulation of the two hypothesis that allowed us to obtain a clear-cut answer.

The statistical analyses appear appropriate. Other specific points for the authors' consideration are listed below:

Major points:

1) Line 83: The authors describe the presence of EVs in the MRS medium. Other than the average size of these EVs, no further information is provided. It is likely that the EVs present in the medium are of eukaryotic origin, however, as these have been used as controls in some experiments, the authors need to characterize them further.

We followed the reviewer's suggestion to characterize the MRS medium. MRS medium used to growth bacteria in our study was autoclaved at 121 °C for 15 min. As we reported in the original version of our manuscript, using NTA technology we found some particles in the MRS media that are distinguished from bacterial EVs by smaller sizes and numbers. Now, to address the reviewer's questions, we lysed EVs derived from MRS medium and bacteria and separated them by SDS-PAGE gel electrophoresis. No specific protein bands were found in MRS-isolated particles in contrast to bacterial EVs. This may be expected, since the MRS medium was autoclaved, which most likely resulted in denaturation of the proteins present in the MRS medium. Moreover, we did not find eukaryotic EV-markers (TSG 101, CD63) in any EVs tested in our study, demonstrating that the preparation of bacteria-derived EVs did not contain eukaryotic EVs (see Fig.1d; Methods section, lines 584-595). We present these results on lines 88-92 and discuss them on lines 319-324.

2) Line 124 Fig. 3): A positive control is required to show that cell death can be detected in this model.

We agree with the reviewer and added a positive control (Triton X-100 at 0.2%) in the amended version of our paper. Besides the two assays described in the original manuscript (propidium-

iodide assay and flow cytometry), we also used an MTT assay to evaluate cell metabolic activity as a proxy for cell viability. As expected, the positive control Triton X-100 at 0.2% clearly reduced cell viability in our assays (see Figs. 3a,b). We present these results on lines 149-150 and lines 156-158.

3) *Lines 139-143 (Fig. 4a): MRS EVs also need to be tested in this cell line, as in Fig. 2a.*

4) *Lines 163-166 (Fig. 5a): MRS EVs need to be tested in this cell line.*

We followed the reviewer's recommendations and tested the effect of EVs from MRS, and we now report the findings in the manuscript, as suggested by the reviewer. As expected, EVs from MRS did not significantly impact either HIV-1 replication or viral entry/attachment. See Fig.4a, Fig.5a, and Fig.5b. We present these results on lines 177-179, 202-204, and 211-215.

5) *Lines 167-172 (Fig. 5b): Do the BC3 EVs also block HIV-1 cell attachment, whereas those of BC5, BC13 and MRS do not?*

As suggested by the reviewer, we performed cell attachment experiments including EVs derived from *L. crispatus* BC3, *L. crispatus* BC5, *L. gasseri* BC13, and MRS. We do see that EVs released from *L. crispatus* BC3, as well as EVs released from *L. gasseri* BC12 significantly decreased HIV-1 cell attachment. In contrast, EVs derived from *L. crispatus* BC5, *L. gasseri* BC13, and MRS did not decrease viral attachment on target cells (see Fig.5b). We present these results on lines 205-215

6) *Lines 185-214: The data from the metabolic and proteomic studies are not convincing.*

*Although a PCA of the metabolite composition of EVs identified some trends between the EVs that blocked HIV-1 entry and those that didn't, the data are based on only 5 different samples, therefore making it impossible to reach any firm conclusions. Similarly, the proteomic analyses have really only identified proteins that are commonly found in bacterial EV preparations. The authors would need to show that the inactivation of one or more of the corresponding genes had an effect on the protective activity of the EVs. Alternatively, they would need to perform proteomic analyses on multiple samples (biological replicates) for each type of EV to show that the trends are reproducible and/or analyze more types of EVs. Related to this point, were multiple batches of EVs for each *Lactobacillus* spp. prepared and tested or were biological replicates of cell culture studies performed with just one batch of each type of EV?*

Following the reviewer's suggestion, we repeated the entire metabolomic investigation on three newly prepared different batches per sample. These three replicates were obtained using a new batch of growth medium (MRS) and are characterized by a slightly different metabolome profile. Therefore, they were analyzed separately from the previous ones. The higher amount of samples allowed us to investigate the Pearson's correlation between the concentrations of each molecule from the metabolomic profile and their effectiveness against HIV-1. This information has been included in the Methods section (lines 597-611). This univariate approach confirmed and refined the original one based on a single repetition per sample. Indeed, the five molecules whose concentrations significantly correlated with their effectiveness in the new analysis followed the same trend as observed in the original rPCA model. In the new analysis, the link between

effectiveness and metabolome appears along PC 1 (accounting for 74.8% of the variance described by the model). We assessed the reproducibility of the biological replicates by calculating intra- and inter-sample variability. We present this results in Fig. 8, lines 253-265.

Our proteomic data presented in Table 1 and in Supplementary Table 2 had already been performed with two different EV batches per sample (run 1, run 2). Moreover, all our experiments were done using freshly extracted EVs or EVs maintained at 4°C for up to four weeks.

We agree with the reviewer that the inactivation of one or more of the corresponding genes could confirm the role of these proteins on HIV-1 inhibition observed in our study. However, experiments on gene silencing are beyond the scope of our study, which aimed to describe the effects of bacterial EVs on HIV-1 infection. Also, we believed that bacterial EVs inhibit HIV-1 at the entry/attachment level. Unless the reviewer strongly disagrees, we would like to keep the proteomic data in our manuscript.

Minor points:

1) *Line 69: Reiterate here the reason here for the focus on vaginal lactobacilli and briefly mention the reason(s) for choosing these particular strains.*

As the reviewer suggested, we reiterate the reason for the focus on vaginal lactobacilli and the reasons for choosing these particular viral strains. In particular, the choice of *Lactobacillus* strains (*L. crispatus* BC3, *L. crispatus* BC5, *L. gasseri* BC12, *L. gasseri* BC13) was based on the earlier reported anti-HIV-1 activity of these bacteria in human tissues *ex vivo* (Ñahui Palomino, 2017). Moreover, these bacteria represent the species that mostly dominate the vaginal ecosystem (Ravel, 2011). We added these points in the amended version of our paper (see lines 62-65).

2) *Lines 90-91: More information is required here for the non-specialist reader.*

As the reviewer suggested, we now provide more information for the non-specialist readers (see lines 98-101).

3) *Line 113: Why were tonsillar tissues used?*

Tonsillar tissue explants together with cervico-vaginal tissue explants are the standard tissue system for the study of HIV-1 pathogenesis as they faithfully reflect many aspects of tissues where critical events in HIV-1 transmission and pathogenesis occur *in vivo*. These tissues support productive HIV-1 infection without exogenous activation or stimulation and retain the pattern of expression of key cell surface molecules relevant to HIV-1 infection (Grivel, 2009). No differences were observed in the effects of EVs in these two experimental systems. We added this information on lines 326-332 of the revised manuscript.

REVIEWERS' COMMENTS:

Reviewer #1 (Remarks to the Author):

The authors have responded to my comments.

Reviewer #2 (Remarks to the Author):

The authors provided satisfactory responses to my concerns that I raised with my initial review. The revised manuscript is improved with more clarity in some of the figures that I requested. They have adjusted their claims with respect to in vivo relevance as I had pointed out as these were in vitro experiments. The paper is a very interesting study with intriguing implications regarding the microbiome in mucosal immunity.

Reviewer #3 (Remarks to the Author):

The authors have satisfactorily addressed the reviewers' comments, resulting in a significantly improved manuscript. Nevertheless, there are a few minor points that need addressing.

1) The authors addressed my question about the potential presence of eukaryotic EVs in the MRS medium by Western blotting using EV markers (lines 88-92). No cross-reactivity was detected with the MRS-derived "EVs" suggesting that the particles were not of eukaryotic origin. However, the authors need to provide a positive control to show that it is not a false negative result.

2) In some places, the authors still refer to the particles in the MRS medium as "EVs" and in other instances, as "particles." (See lines 93, 111, 179, 428, 476, 521, 829 etc and the labelling in many of the figures.) For consistency and clarity, please refer to these throughout as "particles".

3) Change to "stages" (line 100).

4) "Along such PC1..." (lines 257, 264). The meaning of this expression is not clear.

5) Change to "gastrointestinal bacterium, *Lactobacillus casei*."

6) The sentences in lines 392 and 404-405 do not read well.

POINT-BY-POINT RESPONSE TO THE REFEREES' COMMENTS

In our point-by-point response, line numbers are referred to the clean revised version of the uploaded manuscript.

Reviewer #1 (Remarks to the Author):

The authors have responded to my comments.

We thank Reviewer #1 for the positive evaluation of our work.

Reviewer #2 (Remarks to the Author):

The authors provided satisfactory responses to my concerns that I raised with my initial review. The revised manuscript is improved with more clarity in some of the figures that I requested. They have adjusted their claims with respect to in vivo relevance as I had pointed out as these were in vitro experiments. The paper is a very interesting study with intriguing implications regarding the microbiome in mucosal immunity.

We thank Reviewer #2 for the positive evaluation of our work.

Response to Reviewer 3:

Reviewer #3 (Remarks to the Author):

The authors have satisfactorily addressed the reviewers' comments, resulting in a significantly improved manuscript. Nevertheless, there are a few minor points that need addressing.

We thank Reviewer #3 for the positive evaluation of our work and for pointing out the few minor points that needs to be addressed. In the uploaded version of our paper we addressed all these points and amended our manuscript according to the Reviewer's recommendation.

Specifically:

1) The authors addressed my question about the potential presence of eukaryotic EVs in the MRS medium by Western blotting using EV markers (lines 88-92). No cross-reactivity was detected with the MRS-derived "EVs" suggesting that the particles were not of eukaryotic origin. However, the authors need to provide a positive control to show that it is not a false negative result.

We followed the reviewer's request and performed new Western blotting experiments that included a positive control, lysates from mammalian MT-4 cells. Our results confirm the absence of eukaryotic EV-markers in all *Lactobacillus*-derived EVs. Also, we demonstrate that our data are not due to a false negative result as the bands, specific for the antibodies, are present in the positive control. We present these results on Fig. 1d and in lines 89-92.

2) In some places, the authors still refer to the particles in the MRS medium as "EVs" and in other instances, as "particles." (See lines 93, 111, 179, 428, 476, 521, 829 etc and the labelling

in many of the figures.) For consistency and clarity, please refer to these throughout as “particles”.

We have changed for “particles” through the manuscript and figures as reviewer suggested.

3) *Change to “stages” (line 100).*

We corrected the typo to “stages” (see line 100).

4) *“Along such PC1...” (lines 257, 264). The meaning of this expression is not clear.*

Thank you for pointing out this statement. To make it clear, we replaced the sentence in line 257 to the following: “EVs derived from *L. crispatus* BC3 and *L. gasseri* BC12, which efficiently suppress HIV-1 inhibition, showed lower PC 1 scores. EVs from *L. crispatus* BC5 and *L. gasseri* BC13, which did not inhibit HIV-1, showed higher PC 1 scores” (lines 270-273 of the amended version of our paper).

Also, we replaced the sentence in line 264 to the following: “The samples from the same bacterial strains showed similar PC scores, thus suggesting a good biological reproducibility of the metabolome’s profile” (lines 277-278 of the amended version of our paper).

5) *Change to “gastrointestinal bacterium, Lactobacillus casei.”*

We have changed it to “gastrointestinal bacterium, *Lactobacillus casei*”, as the reviewer suggested (see line 379).

6) *The sentences in lines 392 and 404-405 do not read well.*

To make this sentence read better, we corrected the first of these sentences (see lines 405-408) and deleted the second one.